# Potential of Green Ceramics Waste for Alkali Activated Foams

**DOI:** 10.3390/ma12213563

**Published:** 2019-10-30

**Authors:** Barbara Horvat, Vilma Ducman

**Affiliations:** Slovenian National Building and Civil Engineering Institute, Dimičeva ulica 12, Ljubljana 1000, Slovenia

**Keywords:** alkali activation, foaming, SEM, XRF, XRD, mechanical strength

## Abstract

The aim of the paper is to research the influence of foaming and stabilization agents in the alkali activation process of waste green ceramics for future low cost up-cycling into lightweight porous thermal insulating material. Green waste ceramics, which is used in the present article, is a green body residue (non-successful intermediate-product) in the synthesis of technical ceramics for fuses. This residue was alkali activated with Na-water glass and NaOH in theoretically determined ratio based on data from X-ray fluorescence (XRF) and X-ray powder diffraction (XRD) that was set to maximise mechanical properties and to avoid efflorescence. Prepared mixtures were compared to alkali activated material prepared in theoretically less favourable ratios, and tested on the strength and density. Selected mixtures were further foamed with different foaming agents, that are Na-perborate (s), H_2_O_2_ (l), and Al (s), and supported by a stabilization agent, i.e., Na-dodecyl sulphate. The goal of the presented work was to prepare alkali activated foam based on green ceramics with density below 1 kg/l and compressive strength above 1 MPa.

## 1. Introduction

Alkali activated materials (AAM) represent a promising alternative in the building industry for cement, mortar, concrete, pavements, ceramics, etc. because [1,2,3,4,5,6,7,8,9,10,11,12]: AAM’s main ingredient can be inexpensive waste material in powder form containing enough amorphous Al and Si, which form the aluminosilicate network (ASN), when waste material is activated with alkali (usually alkali hydroxide and/or alkali silicate), as one-part (alkali and precursor are mixed in dry state, water is added later [13]), or two-part system (precursor is mixed with liquid alkali).Waste material can be inert or containing small amount of environmentally problematic, hazardous, and even radioactive elements, which get completely immobilized in ASN.With AAM synthesis, i.e., upcycling of waste material contributes to waste-free circular economy.Has potential to lower building industry’s CO2 emission when compared to materials from Portland cement, since (almost) no carbon footprint is attributed to waste precursors.Have lower energy needs for synthesis when compared to products from Portland cement.AAM can have high temperature resistance.AAM can be chemically stable and resistant.

The process of polymerization starts after dissolution: Al atoms penetrate into the Si-O-Si structure, bonds Si-O-Si brake, and new phases arise. The products are mostly alumino-silicate gels with Equation (1):(1)$$M_{n}{\left[-{\left(\mathrm{Si}-O\right)}_{z}-\mathrm{Al}-O\right]}_{n}\cdot {\mathrm{wH}}_{2}O$$
where M is alkali ion, n is the degree of poly-condensation, z, n, and w is 1, 2, 3, or more. Network, usually in the amorphous phase, consists of SiO_4_ and AlO_4_ tetrahedrons that are joined by oxygen bridges, negative charge of Al^3+^ is compensated with positive ions [14].

For alkali activated materials to become lightweight and insulating, foaming agents, such as Na-perborate, Al powder, H_2_O_2_, silica fumes, etc., are added before or in the early stages of alkali activation process. Gases are released in the slurry of precursor and alkali activator due to the chemical reactions of the foaming agent with the liquid network [15].

The mechanisms of hydrogen gas formation with Al powder is given in Equation (2) [15,16,17,18]:$$2{\mathrm{Al}}_{\left(s\right)}+6H_{2}O_{\left(l\right)}\to 2\mathrm{Al}{\left(\mathrm{OH}\right)}_{3\left(\mathrm{aq}\right)}+3H_{2\left(g\right)},$$
$$2{\mathrm{Al}}_{\left(s\right)}+6H_{2}O_{\left(l\right)}+2{\mathrm{NaOH}}_{\left(s\right)}\to 2\mathrm{NaAl}{\left(\mathrm{OH}\right)}_{4\left(\mathrm{aq}\right)}+3H_{2\left(g\right)},$$
(2)$$2{\mathrm{Al}}_{\left(s\right)}+2H_{2}O_{\left(l\right)}+2{\mathrm{NaOH}}_{\left(s\right)}\to 2{\mathrm{NaAlO}}_{2\left(\mathrm{aq}\right)}+3H_{2\left(g\right)}$$

Hydrogen gas forms also with silica fume (can contain a small amount of free Si), given in Equation (3) [19,20,21]:(3)$${\mathrm{Si}}_{\left(s\right)}^{0}+4H_{2}O_{\left(l\right)}\to \mathrm{Si}{\left(\mathrm{OH}\right)}_{4\left(\mathrm{aq}\right)}+2H_{2\left(g\right)}.$$

The oxygen gas formation with H_2_O_2_ is presented in Equation (4) [15,17,22]:(4)$$2H_{2}O_{2\left(l\right)}\to 2H_{2}O_{\left(l\right)}+O_{2\left(g\right)}.$$

Oxygen gas forms also in the reaction of Na-perborate with water. Hydrolysis is bimodal and is presented in Equation (5) [22,23]:$${\mathrm{Na}}_{2}^{+}{\left[{\left(B{\left(\mathrm{OH}\right)}_{2}\mathrm{OO}\right)}_{2}\right]}_{2\left(s\right)}^{-}+2H_{2}O_{\left(l\right)}⇌2{\mathrm{Na}}^{+}+2{\left[B{\left(\mathrm{OH}\right)}_{3}\left(\mathrm{OOH}\right)\right]}^{-},$$
$$[{\left(B{\left(\mathrm{OH}\right)}_{3}\left(\mathrm{OOH}\right)\right]}^{-}⇌B{\left(\mathrm{OH}\right)}_{3}+{\mathrm{HOO}}^{-},$$
(5)$$B{\left(\mathrm{OH}\right)}_{3}+{\mathrm{HOO}}^{-}+H_{2}O⇌{\left[B{\left(\mathrm{OH}\right)}_{4}\right]}^{-}+H_{2}O_{2}$$

Extensive research has been performed in different systems with different foaming agents, were for [15,16,17,18,20,24,25,26,27,28,29,30]: Precursor fly ash, or metakaolin with fly ash, or blast furnace slag, or blast furnace slag with calcined reservoir sludge, or metakaolin were mainly used,Activators Na-water glass and NaOH were used by majority, while some just used NaOH, or NaAlO_2_ and NaOH, or K-water glass and KOH,Foaming agents Al, H_2_O_2_ were used, while there is also some research done with using Na-perborate, Micro Air 210, oleic acid, Na-lauryl sulfate, silica fume were used for foaming agents; while there was research performed also with mechanical foaming,Stabilizing agent Na-dodecyl-sulphate, Sika® Lightcrete 02 were used.

Abdollahnejad et al. [24] gained using fly ash-based geopolymer with H_2_O_2_ or Na-perborate as foaming agents compressive strength from 4.5 to 6 MPa, where the lowest thermal conductivity was reached with Na-perborate, around 0.1 W/(m·K), and the lowest density with H_2_O_2_, i.e., 0.734 kg/l. In principle, they found out, that more foaming agent results in lower density, but could not find correlation between thermal conductivity and density. 

Sanjayan et al. [25] substituted part of fly ash with Al and got lowest density, 0.403 kg/l. Compressive strength ranged from 0.9 to 4.35 MPa. They found out that if the foaming reaction is too fast, it prevents complete alkali activation of fly ash. The liquid part of alkali activator affected density of foamed alkali activated materials severely, i.e., lower density was achieved with more liquid content. Compressive strength decreased with lowering density. Hajimohammai et al. [26] also followed foaming of fly ash with Al powder, where they found out that the higher oxidation rate, the higher rate of foaming, the higher porosity, the lower density, less unreacted fly ash, lower Si: Al ratio in the ASN at later stages of reaction leading to loss of compactness. However, results of density and compressive strength, mentioned in the article, were not presented. The same author found out in followed research [16] that in the beginning Al-hydroxide gel precipitates on fly ash, conceals its reactive surface, lowers its dissolution rate, delays strength development (from more than 15 MPa after seven days to more than 30 MPa after 28 days). Due to Al the unreacted particles are better connected, microstructure is better developed, and part of the alkali activated ASN does not carbonate (less thermonatrite develops in samples which had Al). Hlaváček et al. [18] also researched foaming of alkali activated fly ash with Al. They reported that in pastes with low viscosity pore nucleation is fast, leading to deflation after all gas leaves the sample. If viscosity of the paste was high, pore propagation did not take place and inflation of samples was low. Amount of Al had lesser effect on pore structure than the amount of liquid. Bulk density of optimized alkali activated foam was 0.671 kg/l, compressive strength 6.0 MPa, thermal conductivity 0.145 W/(m·K), thermal capacity 1089 J/(kg·K), high fire resistance (up to 1100 °C), where material sintered and provided long-term chemical durability in several aggressive environments (due to closed pores and no leachable Ca). Masi et al. [27] researched foaming of fly ash with Al and H_2_O_2_, with the addition of surfactant Sika® Lightcrete 02. With surfactant, and with both foaming agents, the compressive strength dropped. Density when only the surfactant was used did not drop below 1.18 kg/l, when Al was used the lowest density obtained was 0.80 kg/l and pore size from 500 to 3500 µm, when only H_2_O_2_ was used as foaming agent the lowest density of 0.91 kg/l with pores from 200 to 3000 µm was reached. Mixture of H_2_O_2_ and surfactant resulted in foams with lowest density, 0.94 kg/l, and compressive strength 4.6 MPa. If foaming agents were used without surfactant, pores were larger than 200 µm. The authors proposed to pay attention to the side effects of foaming agents, i.e., they could take part in the alkali activation process (Al), or additional water formed when the foaming agent used is H_2_O_2_, pH of the slurry lowers, and reduces the alkali activation process. In addition, Ducman et al. [15] researched foaming of alkali activated fly ash with Al powder and H_2_O_2_. With Al they reached up to 59% porosity, with H_2_O_2_ 48%. The lowest density reached with Al powder was 0.64 kg/l, with compressive strength 4.3 MPa, and with H_2_O_2_ 0.61 kg/l, with compressive strength 2.9 MPa. With H_2_O_2_ as foaming agent they got pores with diameter up to 500 µm, with Al some pores’ diameter was bigger than 1 mm. Korat et al. [17] followed the influence of Na-dodecysulphate as stabilizing agent on porosity of fly ash foamed with H_2_O_2_. They noticed that density drops with increasing amount of stabilizing agent, as well as with increasing amount of foaming agent. The larger quantity of foaming agent had a significant effect on pore size distribution. The lowest reached density was 0.58 kg/l, and highest total porosity 54.5%. Novais et al. [28] researched foaming of mixture of metakaolin and fly ash with H_2_O_2_. Compressive strength generally grows with curing time, except in two mixtures where decrease happened in the first seven days, where they suggest that phase changes are taking place. Highest compressive strength, 21 MPa, was achieved without foaming agent. Increasing the amount of water decreases compressive strength, even a bigger decrease of compressive strength occurred with increasing amount of H_2_O_2_. Lowest compressive strength reached was 0.26 MPa, that was accompanied by lowest density, 0.44 kg/l. Esmaily et al. [29] investigated the possibility to replace autoclaving curing stage by steam curing while replacing cementious materials with alkali activated blast furnace slag, which was foamed with Al powder and with additional foaming agents: Micro Air 210, oleic acid, and Na-lauryl sulfate. Ratio SiO_2_:Na_2_O is according to their conclusions one of the most effective parameters in compressive strength of the alkali activated slag cement, that also hinders/allows reaction of Al with low/high basicity. Na-lauryl sulfate showed as the most promising foaming agent, with compressive strength 7.8 MPa, at viscosity of the pastes that were for all foaming agents the same, but all the rest had compressive strength below 1 MPa. They found out that compressive strength is higher, if the molds are sealed and curing temperature is higher. Yang et al. [30] researched mechanical foaming of calcined reservoir sludge mixed with 30% of blast furnace slag with alkaline solutions. Highest compressive strength gained was over 60 MPa. Viscosity increased with time, and it had to be less than 5000 mPa·s for the first 20 min. They designed densities of 0.7, 0.8, 0.9, and 1 kg/l with air bubbles being randomly and uniformly distributed. Foams with higher densities had higher bending and higher compressive strengths. Compressive strength was approximately 10-times higher than bending strength. Both strengths grew with time, compressive strength of 0.7 kg/l foam reaching more than 6 MPa after 91 days. Delair et al. [20] researched foaming of metakaolin with silica fume. They noticed that the sample was not homogeneous, i.e., surface and interior of the cured samples did not have equal surface exchange and exposure to temperature, i.e., drying was not uniform. With test of stability in aqueous media they found that no matter which cation of the first group as in the alkali activator, the final pH of the equilibrium between solution and alkali activated foam immersed in aqueous media was the same. Aging resulted in increase of durability (when KOH and K-water glass was used as alkali activator) and in increase in carbonation (when NaOH and Na-water glass was used as alkali activator). The aim of the present research is to valorize the green ceramic waste for its use in the alkali activation process and foaming. Green waste ceramic material is a residue (non-successful intermediate-product/green body, and green scraps from green body) in the synthesis of technical ceramics for fuses. It represents approximately 45% of all residues of waste generated in ceramic tile processes. More than 80% of this waste is mixed with raw materials and used again; remaining is used as fillers in other materials or landfilled [31].

Material is non-dangerous, insoluble in water, non-combustible, already homogeneous, which can be easily grinded into powdered form. Based on data from X-ray fluorescence (XRF) and X-ray powder diffraction (XRD), the optimal ratio of precursors to activator has been calculated in order to reach optimal mechanical properties and to avoid efflorescence. After selection of optimal mixtures they were further foamed by adding different foaming agents such as Na-perborate_(s)_, H_2_O_2(l)_, and Al_(s)_. Their influence of final properties, such as density and compressive strength were assessed.

## 2. Materials and Methods 

### 2.1. Characterization 

Waste green ceramics, labelled 10 12 01 (according the Classification list of waste from Official Gazette of the Republic of Slovenia, no. 20/01 Annex 1, absorbed into Slovenian legislation from European Waste Catalogue and Hazardous Waste List), is residue (unfired reject) from the production of ceramic fuse. Material was taken at company Termit’ s open waste dump (Drtija, Slovenia) in a manner to be representative for the waste dump pile. Sample was dried at 70 °C for 24 h, ground, and sieved.

For X-ray fluorescence (XRF) and X-ray powder diffraction (XRD) analysis waste green ceramics was dried on 70 °C for one day in a drying chamber, then milled and sieved below 90 µm.

XRD analysis was performed on precursor and alkali activated material (Empyrean PANalytical X-Ray Diffractometer, Cu X-Ray source, 45 kV, 40 MA, Thermo Scientific, Thermo electron SA, Ecublens, Switzerland) from 4 to 70° angle in step 0.0263°, under cleanroom conditions, in powder sample holders with aperture diameter 27 mm. From XRD peaks crystallites’ sizes present in waste green ceramics were determined by means of Scherrer formula [32], mass percentage amounts of amorphous phase and minerals were estimated with Rietveld refinement [33] using external standard (pure crystal of Al_2_O_3_) with program X’Pert Highscore plus 4.1 (version 4.1, Malvern Panalytical, Surrey, United Kingdom).

XRF analysis was performed on precursor (Thermo Scientific ARL Perform’X Sequential XRF, 60 kV, 40 mA, Thermo electron SA, Ecublens, Switzerland) with program OXAS on melted disks, which were prepared by mixing sample and Fluxana (FX-X50-2, Lithium tetraborate 50% / Lithium metaborate 50%). Data were characterized with program UniQuant 5. 

Scanning electron microscopy (SEM; Jeol JSM-5500LV, Jeol, Tokyo, Japan) investigation was performed under low vacuum conditions on dried precursor and alkali activated material without coating. 

Compressive and bending strength of alkali activated materials were tested by means of compressive and bending strength testing machine (ToniTechnik ToniNORM, Berlin, Germany) one week after synthesis, if not stated otherwise. 

Density of alkali activated materials was determined by means of geometrical methods, i.e., by weighting of the sample and dividing its mass with geometrically determined volume.

### 2.2. Preparation of Alkali Activated Samples

Waste green ceramics, i.e., green body and green scraps from green body, were dried at 70 °C for one day in drying chamber, then crushed and grinded, and sieved below 1 mm.

Alkali activation was performed by addition of NaOH (Donau Chemie Ätznatron Schuppen, EINECS 215-785-5, Vienna, Austria) to Na-water glass (Geosil, 344/7, Woelner, Ludwigshafen, Germany, with mass percentage of Na_2_O 16.9%, and mass percentage of SiO_2_ 27.5%) or water in different ratios (presented in Table 1), stirred until liquid became clear, poured into the sample in different ratios (presented in Table 1) under constant mixing for 1 min. Viscosity of liquids was measured with viscometer Haake VT 500 with detector NV at 25 °C (presented in Table 1). 

Mass ratio sample: NaOH:Na-water glass of needed optimal value of Na and Si was determined from XRF and XRD analysis according to amount of Al in amorphous phase and ideal ratio of amount of substance n_Al_:n_Si_:n_Na_ = 1:1.9:1 [34] (evaluated on AAM from fly ash), which was set to avoid efflorescence and have highest compressive strength (both amount of substance ratios used in calculations are presented in Equations (6) and (7) and set to be rational numbers in calculations, to be able to change them in the program for the estimation of the additives to the precursor).
(6)$$n_{\mathrm{Si}}:n_{\mathrm{Al}}=\alpha_{1}=1.9,$$
(7)$$n_{{\mathrm{Cation}}^{1+}}:n_{\mathrm{Al}}=\alpha_{2}=1,$$

With a difference to article [34], instead of using whole (XRF) amounts of Al, Si, Na present in the precursor and additives, only the amorphous phase was used, and besides only using Na, the whole first group was taken into account, the second group only for potential further improvement (Equation (8)).
(8)$$\frac{1}{2}n_{{\mathrm{Cation}}^{2+}}:n_{\mathrm{Al}}=\alpha_{2}=1$$

XRF results were recalculated into mass percentage m_%_ of elements (Equation (9)), where A represents an element detected in the precursor, M molar mass (of element A and its oxide A_x_O_y_), and P precursor.
(9)$$m_{{\%}_{P}}{}^{\mathrm{XRF}}\left(A\right)=x\cdot m_{{\%}_{P}}{}^{\mathrm{XRF}}\left(A_{x}O_{y}\right)\cdot \frac{M\left(A\right)}{M\left(A_{x}O_{y}\right)}$$

Mass percentage of elements in crystal form estimated with Rietveld refinement from XRD measurement of the precursor was also recalculated into mass percentage m_%_ of elements (Equation (10)), where B represents an element in crystal form, M molar mass (of element B and mineral B_z_), summed over all minerals containing element B.
(10)$$m_{{\%}_{P}}{}^{\mathrm{XRD}}\left(B\right)=\sum_{\mathrm{minerals}\;“B”}z\cdot m_{{\%}_{P}}{}^{\mathrm{XRD}}\left(B_{z}\ldots \right)\cdot \frac{M\left(B\right)}{M\left(B_{z}\ldots \right)}$$

The amount of amorphous phase, useful in the alkali activated reaction, was calculated as the difference between XRF and XRD mass percentage (Equation (11)).
(11)$$m_{{\%}_{P}}{}^{\mathrm{amorphous}}\left(A\right)=m_{{\%}_{P}}{}^{\mathrm{XRF}}\left(A\right)-m_{{\%}_{P}}{}^{\mathrm{XRD}}\left(A\right)$$

According to the amount of Al present in the precursor and chosen ratio of amount of substance of Al and Si (Equation (6)), the mass of additional needed Si was determined with Equation (12), where m_0_ is mass of the precursor.

(12)$$m_{{\mathrm{Si}}_{\mathrm{add}}}=\frac{m_{0}}{100}\cdot \left(\alpha_{1}\cdot m_{{\%}_{P}}\left(\mathrm{Al}\right)\cdot \frac{M\left(\mathrm{Si}\right)}{M\left(\mathrm{Al}\right)}-m_{{\%}_{P}}\left(\mathrm{Si}\right)\right)=\{\begin{array}{c} \mathrm{true},\;\mathrm{if}>0;\;\mathrm{do}\;\mathrm{not}\;\mathrm{add}\;\mathrm{Al}\\ 0,\;\mathrm{if}=0;\;\mathrm{do}\;\mathrm{not}\;\mathrm{add}\;\mathrm{Al}\\ 0,\;\mathrm{if}<0;\;\mathrm{add}\;\mathrm{Al} \end{array}.$$

If there was more Si than needed, the mass of additional Al, from compound not containing any other crucial element for alkali activation and ASN formation was determined with Equation (13).

(13)$$m_{{\mathrm{Al}}_{\mathrm{add}}}=\frac{m_{0}}{100}\cdot \left(\frac{1}{\alpha_{1}}\cdot m_{{\%}_{P}}\left(\mathrm{Si}\right)\cdot \frac{M\left(\mathrm{Al}\right)}{M\left(\mathrm{Si}\right)}-m_{{\%}_{P}}\left(\mathrm{Al}\right)\right).$$

Minimal needed mass of Na, determined with Equation (14) where the sum goes over all cations from the first and second group present in the amorphous phase, was calculated using Equation (7) and Equation (8).

(14)$$\begin{array}{c} m_{{\mathrm{Na}}_{\mathrm{add}}}=\frac{M\left(\mathrm{Na}\right)}{M\left(\mathrm{Al}\right)}\cdot \left(\frac{m_{0}}{100}\cdot m_{{\%}_{P}}\left(\mathrm{Al}\right)+m_{{\mathrm{Al}}_{\mathrm{add}}}\right)\cdot \alpha_{2}-\frac{m_{0}}{100}\cdot [\sum_{{\mathrm{Cation}}^{1+}}(\frac{M\left(\mathrm{Na}\right)}{M\left({\mathrm{Cation}}^{1+}\right)}\cdot \\ m_{{\%}_{P}}\left({\mathrm{Cation}}^{1+}\right))+\sum_{{\mathrm{Cation}}^{2+}}\left(2\cdot \frac{M\left(\mathrm{Na}\right)}{M\left({\mathrm{Cation}}^{2+}\right)}\cdot m_{{\%}_{P}}\left({\mathrm{Cation}}^{2+}\right)\right)]. \end{array}$$

If there was not enough added Na to compensate the negative charge of Al in the ASN, the mass of needed K, determined with Equation (15), was determined using Equation (7) and Equation (8), and already added mass of Na.

(15)$$\begin{array}{c} m_{K_{\mathrm{add}}}=\frac{M\left(K\right)}{M\left(\mathrm{Al}\right)}\cdot \left(\frac{m_{0}}{100}\cdot m_{{\%}_{P}}\left(\mathrm{Al}\right)+m_{{\mathrm{Al}}_{\mathrm{add}}}\right)\cdot \alpha_{2}-\frac{M\left(K\right)}{M\left(\mathrm{Na}\right)}\cdot m_{{\mathrm{Na}}_{\mathrm{add}}}-\frac{m_{0}}{100}\cdot \\ {}[\sum_{{\mathrm{Cation}}^{1+}}\left(\frac{M\left(K\right)}{M\left({\mathrm{Cation}}^{1+}\right)}\cdot m_{{\%}_{P}}\left({\mathrm{Cation}}^{1+}\right)\right)+\sum_{{\mathrm{Cation}}^{2+}}\left(2\cdot \frac{M\left(K\right)}{M\left({\mathrm{Cation}}^{2+}\right)}\cdot m_{{\%}_{P}}\left({\mathrm{Cation}}^{2+}\right)\right)]. \end{array}$$

Minimal needed liquid phase (distilled water, including water from Na-water glass) was experimentally determined, i.e., liquid phase had to wet the whole precursor’s surface and viscosity had to be too high to be measured. The amount of liquid phase was further varied no matter the efflorescence and viscosity.

Alkali activated (AA) mixtures were put into molds of sizes (80 × 20 × 20) mm^3^ and left to dry and solidify on 70 and 90 °C for one day if not stated otherwise. 

### 2.3. Preparation of Alkali Activated Foams

Alkali activation foams were prepared by adding different foaming agents: Na-perborate monohydrate (powder, Belinka Perkemija d.o.o, Ljubljana, Slovenia), hydrogen peroxide solution 30% (liquid, H_2_O_2_, Carlo Erba reagents, Barcelona, Spain), and aluminium (powder, Particle Size D50 < 5 µm, NMO, New materials development Gmbh, Heemsen, Germany); and stabilizer: Na-dodecyl sulphate (powder, Acros Organics, Geel, Belgium), in different mass ratios to precursor presented in Table 2—mass ratios varied for different foaming agents as can be seen in the results Section 3.4 (liquid phase is presented in Table 1), i.e., if the foaming agent produced very high foaming effect, the amount of foaming agent was lowered. Powder additives were added directly to the dry precursor, well mixed, and afterwards the liquid phase needed for alkali activation was added. On the other hand, liquid additives were directly added to the alkali activating liquid phase. 

## 3. Results and Discussion

### 3.1. Analysis of Precursor

Surface and shape of green waste ceramics is presented on SEM images in Figure 1. Precursor consists of particles’ sizes generally below 100 µm, and with sharp edges and smooth surface.

The amount of different oxides was followed by means of XRF. Results for Al, Si, first and second group of the periodic system are presented in the first row in Table 3. The amount of amorphous phase and minerals was determined with Rietveld refinement (with goodness of fit 8.8 due to the precursor being waste material with few indefinable small peaks, meaning there is less amorphous phase than determined; presented in Figure 2), and recalculated onto elements presented in the second row in Table 1. Amount of elements useful in alkali activation, i.e., elements that are in the amorphous phase, presented in the third row in Table 3, was calculated by rational analysis as difference between whole amount of elements (XRF) and elements present in minerals (XRD), excluding the possibility of minerals dissolving in alkali, as presented in Section 2.2.

Ratio of all elements in waste green ceramics is n_1st group_:n_Al_:n_Si_ = 0.23:1:2.26, which when compared to the ratio n_1st group_:n_Al_:n_Si_ = 1:1:1.9 [34] means that there is too much Si or not enough Al, and not enough Na, while the ratio of amorphous elements is n_1st group_:n_Al_:n_Si_ = 0.34:1:1.63, meaning Si and Na have to be added; Na-water glass was used as source of Si and Na, and NaOH was used as source of Na.

Difference in ratios n_1st group_:n_Al_:n_Si_ when only amorphous (XRF-XRD) or when all elements (XRF) are taken into account is highly important in noticeably crystalline samples, in order to prepare the correct recipe.

### 3.2. Calculation of Mixtures Based on Data from 2.2 and 3.1

From the available literature ratio (n_1st group_:n_Al_:n_Si_ = 1:1:1.9) should give the highest compressive strength for fly ash [34], and taking into account the whole first group of the periodic system and not just Na, four different mixtures were prepared, where we set n_1st group_:n_Al_ = 1–1.2 and n_Si_:n_Al_ = 1.9–2.5 (in Table 4 rows close to the literature’s optimal value for n_1st group_:n_Al_:n_Si_ are coloured orange). All other performed experiments have higher values and are prone to efflorescence and growth of crystals containing Na. For comparison also ratios of the initial values of elements present in the precursor (grey coloured row in Table 4) are given.

### 3.3. Analysis of Bulk AAM

#### 3.3.1. Dependence on Curing Temperature and Amount of Substance Ratio First Group: Al:Si on Limit Value of Compressive Strength

In Table 5 are presented measured limit values of compressive strength performed on alkali activated samples prepared with 50 g of mixtures ζ_0_, ζ_2_, and ζ_5_, cured on room temperature and later at 70, at 70, and 90 °C. In the same table are presented determined amounts of AAM, H_2_O, ASN and amount of substance ratio of elements important for AA. Rietveld refinement analysis (goodness of fit (GOF) ranges from 7.2 to 8.8) and determination of crystallite size using Scherrer equation., compared with the precursor, are presented in Table 6 (amount of minerals and amorphous phase) and Table 7 (crystallites’ sizes). From the results presented in Table 5 to Table 7 it can be concluded:
Calcite, magnesioferrite, and ankerite completely dissolve in alkali media.Quartz partially dissolves in alkali media, especially on higher temperature (90 °C), remaining crystallites grow, a bit more on milder temperature conditions, and more if the amount of added/whole H_2_O was smaller, i.e., mixture had higher viscosity, i.e., for mass of added/whole H_2_O and viscosity of liquid, size of crystallites it can be written as ζ_0_ > ζ_5_ > ζ_2_, while for final m_%_ of quartz it is vice versa ζ_0_ < ζ_5_ < ζ_2_.Corundum crystallites additionally form, but are smaller compared to the corundum in the precursor. The amount of corundum is bigger when amount of added/whole H_2_O is higher and viscosity is lower, i.e., m_%_ of corundum grows as ζ_0_ < ζ_5_ < ζ_2_, which is vice versa to quartz.New minerals are formed at certain conditions with alkali activation: Microcline, orthoclase, zirconia, kaolinite, zeolite, albite, and sodalite.
⚬Albite (heat treated) grows only when curing was performed on 90 °C. This is the only mineral containing Na that grows also when mixture ζ_2_ is used, while when mixtures ζ_0_ and ζ_5_ are used, also other minerals containing Na grow, i.e., these two mixtures have excessive amount of Na.⚬Microcline grows only when mixture ζ_0_ is used, at 70 °C with a bit of Na, at 90 °C five-times bigger amount and five-times smaller crystallites just with K from the first group.⚬Orthoclase is the only mineral that formed at all temperatures and with ζ_0_, ζ_2_, and ζ_5_.⚬Kaolinite, the softest mineral according to Mohs scale that is present in XRD results of tested AAM, is not present only when mixture ζ_2_ is used. Small amount of kaolinite (less than 2%) in the form of bigger crystallites (too big for correct estimation of crystallites’ size with X’Pert Highscore software) is present in AAM prepared with mixture ζ_5_ at 90 °C with compressive strength above 30 MPa, while at 70 °C there is more than 10% in the form of 15 nm big crystallites and compressive strength around 2.5 MPa. Samples prepared with ζ_0_ have a bit less than 10% of kaolinite with crystallites’ size 20 nm and compressive strength around 10 MPa. Explanation for big difference in compressive strengths’ value, if we make the assumption of uniform distribution of minerals in the prism, is that for bigger amount of smaller crystallites of kaolinite there are more weakest spots through the whole prism, while for smaller amount of bigger sized crystallites there are just few weakest spots in the prism. 

From the amount of amorphous phase present in AAM (presented in Table 7) it can be concluded:When mixture ζ_0_ is used, the temperature has no obvious impact on the amount of amorphous phase probably due to the lowest amount of H_2_O present in the precursor, i.e., due to the highest viscosity that hinders speed of reaction (and evaporation of H_2_O) and gives time for formation and growth of minerals. However, temperature has an impact on which minerals form; When mixture ζ_2_ is used at lowest temperature conditions (room temperature for 4 h followed by curing on 70 **°**C for 20 h), the ions have more time to arrange into positions with lowest energy (slower evaporation of H_2_O) state compared to preparation of AAM just at 70 **°**C, i.e., amount of amorphous phase is higher at curing at just 70 **°**C. When curing was performed at 90 **°**C albite (heat treated) formed (in all XRD analyzed experiments) with which the amount of amorphous phase lowered and was comparable to AAM prepared at lowest temperature conditions. However, excluding albite (heat treated) temperature did not have significant impact on mineral formation and growth.When mixture ζ_5_ is used at 70 **°**C ions have more time for formation and growth of minerals compared to synthesis at 90 **°**C where reaction ends faster (due to forced drying with higher temperature) and leaves material with more amorphous phase. Temperature does show impact on mineral formation, i.e., at lower temperature there was much more kaolinite, at higher temperature there was also albite (heat treated).

#### 3.3.2. Dependence of Mechanical Properties on Amount of Liquid Phase and Amount of Substance Ratio First Group: Al:Si

Compressive strength, bending strength, and density of alkali activated green ceramics, prepared only using Na-water glass (mixture ζ_0_) and mass ratio NaOH: Na-water glass: H_2_O = 4:20:26 (mixture ζ_4_), measured one week after molding is presented in Figure 3 and Table 8. 

Density of AAM prepared with using only Na-water glass (mixture ζ_0_) was slightly higher when compared to AAM prepared with mixture ζ_4_ when the same amount of liquid phase was used, which could be due to moisture value of one week old AAM, which was in the case of mixture ζ_0_ between 8% and 10%, in the case of AAM prepared with mixture ζ_4_ around 5%. In both cases density did not show high dependence on the amount of added liquid phase, but was slightly higher with higher percentage of the precursor. The values of density were from 1.7 to 2.1 kg/l.

Bending strength dropped downwards with the amount of liquid, Na, and Si in one week old AAM prepared only using Na-water glass (mixture ζ_0_), while bending strength of AAM prepared with mixture ζ_4_ was approximately constant. 

Drop downwards of the value of the bending strength of AAM prepared with mixture ζ_0_ could be explained again with higher moisture when AAM was prepared with higher amount of Na-water glass (measured on day seven when AAM still did not reach its limit values; see Section 3.3.3.). Highest value of bending strength, 8.5 MPa, was with the lowest amount of Na-water glass used in AA (25 g), which was one of the four favourable experiments presented in Section 3.2 (orange rows in Table 4). 

Highest value of bending strength when mixture ζ_4_ was used, was 5.5 MPa, prepared with 30 g of mixture ζ_4_, which is also one of the hypothetically favourable experiments presented in Section 3.2 (orange rows in Table 4). Slight drop downwards in bending strength value with higher amount of mixture ζ_4_ starts after the amount of substance ratio between the first group and Al is 1.4 (40 g of mixture ζ_4_).

Compressive strength of mixture ζ_0_ dropped downwards with increased amount of liquid, Na, and Si from 17.6 to 0 MPa (measured on day seven when AAM still did not reach its limit values), while when for the AAM mixture ζ_4_ was used, the drop downwards with amount of liquid started after hypothetical favourable amount of substance ratio between the first group and Al (orange rows in Table 4), i.e., after 30 g of mixture ζ_4_. The drop downwards in compressive strength could be a consequence of the increase of amount of Na rather than moisture, because H_2_O has lower viscosity than Na-water glass, therefore the reactions and drying finish sooner due to the easier diffusion in the material.

From these results it can be concluded that AA mixtures that are closer to the proposed amount of substance ratio of the first group, Al and Si from literature [34] are, as expected favourable according to compressive strength values.

#### 3.3.3. Time Dependence of Mechanical Properties on Curing Temperature and Amount of Substance Ratio First Group: Al:Si

Time dependence of density, bending, and compressive strength were followed for one mixture, worst case in the previous experiment, when AAM was prepared with 50 g of mixture ζ_0_ (ζ_0_^50^), i.e., with amount of substance ratio n_1st group_:n_Al_:n_Si_ = 2.1:1:3.1. Results are presented in Table 9 and Figure 4 (limit values of compressive strength are presented also in Table 5). Since strength values when the sample was cured on 70 °C for one day were not measurable (0 MPa, see Table 8 and Figure 3), curing time on 70 °C was prolonged from one to seven days, in a second set of experiments curing was performed on 90 °C for one day. When curing temperature is higher, compressive and bending strength reach measurable value sooner compared to AAM cured at lower temperature due to accelerated solidification and drying, however, the limit value of compressive strength is higher when curing is performed at lower temperature presumably due to the creation of less deformations/cracks in the AAM structure while drying. In any case, the limit value is lower compared to AAM prepared in hypothetically best amount of substance ratio of the first group, Al, and Si [34] prepared from mixture ζ_0_ and mixture ζ_4_ at 70 °C (see Table 8 and Figure 3).

Observed time dependence of the increase of compressive strength value ${\mathrm{CS}}_{↑}$ can be described with the rational function, presented in Equation (16),
(16)$$CS_{↑}\left(t\right)=\frac{a\cdot \left(t-t_{0}\right)}{\sqrt{t^{2}+b\cdot t+c}}$$
where t is time, a is compressive strength limit value, t_0_ is time when compressive strength was still zero, b and c are parameters in the first approximation including all events taking place in AAM synthesis that are contributing to the compressive strength value, where c>0,$\;b>-\left(\frac{c}{t}+t\right)$ for all t.

Bending strength was higher if the sample was cured at lower temperature, while density was comparable for AAM cured on lower and higher temperatures (see Table 9 and Figure 4).

Time dependence of density, bending, and compressive strength for AAM prepared with 50 g of mixture ζ_2_, i.e., with favourable amount of substance ratio according to the literature [34] n_1st group_:n_Al_:n_Si_ = 1:1:1.9, are presented in Table 10 and Figure 5 (limit values of compressive strength are presented also in Table 5). Mixture ζ_2_ contains the highest amount of water among all experiments, which is a presumed reason that majority of prepared prisms were cracked on halves, therefore majority of bending strengths could not be measured. However, samples reached limit values of compressive strength soon with only potential chemical and physical effects taking place on smaller scale due to realistic distribution of ingredients (not completely uniform). There was no relevant difference noticed between different curing temperatures, the compressive strength limit was basically the same for all three sets of experiments. Density of AAM prepared on higher temperature is always a bit smaller than density of AAM prepared at lower temperatures, while density of AAM prepared on 70 °C with and without waiting on room temperature, were comparable.

Time dependence of density, bending, and compressive strength for AAM prepared with 50 g of mixture ζ_5_ are presented in Table 11 and Figure 6 (limit values of compressive strength are presented also in Table 5). Compressive and bending strengths are lower for AAM prepared on 70 °C compared to AAM prepared on 90 °C for all times. Compressive strength when AAM was prepared on 70 °C slowly grows with time with Equation (16), while sample prepared on 90 °C showed not expected behaviour, which could not be part of the error. Therefore, the experiment was repeated with which high compressive strength and falling time dependence of compressive strength was confirmed two-times. Difference in value of the compressive strength is explained in Section 3.3.1.

Higher initial compressive strength when AAM with mixture ζ_5_ was prepared on higher 90 °C could be due to the formation of minerals of bigger Mohs hardness value, which have to be less stable and transform with time into more stable minerals with smaller hardness. Possible effect for lowering of the compressive strength could be also efflorescence of Na salts. The drop downwards of compressive strength can be described with Equation (17),
(17)$$CS_{↓}\left(t\right)=CT^{\prime }-\frac{a^{\prime }\cdot \left(t-t_{0}{}^{\prime }\right)}{\sqrt{t^{2}+b^{\prime }\cdot t+c^{\prime }}}$$
where $CS_{↓}$ is the compressive strength falling with time t to limit value $\left(CT^{\prime }-a^{\prime }\right)$, $t_{0}{}^{\prime }$ is the time when compressive strength was highest, $b^{\prime }$ and $c^{\prime }$ are parameters including all events taking place in AAM synthesis, where $c^{\prime }>0$, $b^{\prime }>-\left(\frac{c^{\prime }}{t}+t\right)\;$for all t.

Lowest compressive strength among all “time-experiments” cured for 24 h has AAM which was prepared with 50 g of mixture ζ_5_ and cured on 70 °C for 24 h. Possible reason for that was the higher level of moisture (close to 10%), i.e., just like when AAM with 50 g of mixture ζ_0_ was prepared, curing on 70 °C for 24 h was not enough to gain measurable compressive strength, i.e., when prisms from both samples were compressed, AAM did not break, it just started to squeeze. 

Therefore, if the need is to gain measurable compressive strength in short time, the temperature of curing has to be higher, if the used liquid phase has higher viscosity. However, for the limit value of the compressive strength hypothetical relevant parameters could be:Temperature of curing: Lower temperature results in fewer deformations in AAM, which are a consequence of physical effects—evaporation of liquid (drying of AAM); at certain temperature minerals of higher/lower hardness form-degrade, which contribute to the compressive strength value of AAM (see AAM prepared with 50 g of mixture ζ_5_ cured on 70 and 90 °C in Table 6 where 10% of kaolinite, mineral with lowest Mohs hardness present in the sample, form at 70 °C and just 1.5% at 90 °C),time of curing: Too short time of curing on selected temperature can result in not finite AAM, i.e., the chemical and physical reactions are still taking place (see Figure 3a, Table 9 and Figure 4 where samples prepared with 50 g of mixture ζ_0_ on 70 °C do not reach measurable compressive strength in few days),chemistry of the selected precursor and needed additives: If the amount of additives is correct, compressive strength reaches final value soon probably due to the chemical reactions taking place mostly in the beginning of the AA (see Table 10 and Figure 5 where samples prepared with 50 g of mixture ζ_2_ reach limit value soon and are comparable for all curing temperatures), as well efflorescence and formation of certain minerals is avoided (see Table 6 where samples prepared with 50 g of mixture ζ_2_ on 70 °C did not form any mineral containing Na, while in samples with mixtures ζ_0_ and ζ_5_ they did; only samples prepared with 50 g of mixture ζ_2_ did not form kaolinite, i.e., mineral with lowest Mohs hardness are present in the tested samples),amount and viscosity of liquid phase: Too much water can lead to cracked AAM (see Table 10 where majority of samples prepared with 50 g of mixture ζ_2_ do not have result of the bending strength due to prisms being cracked), too high viscosity to very time-consuming reaction (see Figure 3a, Table 9 and Figure 4 where samples prepared with 50 g of mixture ζ_0_ on 70 °C do not reach measurable compressive strength in few days). 

#### 3.3.4. Dependence of Mechanical Properties on Amount of Substance Ratio First Group: Al:Si

In Figure 7 and Table 12 is presented dependence of compressive strength and density on the amount of substance ratio of the first group, Al and Si (presented in Figure 8a), where amount of liquid phase was kept constant, but the chemistry, amount of water, and viscosity of the used liquid changed due to the different mixtures. While density did not show big impact on parameters that were varied, compressive strength had highest value when the amount of substance ratio of the first group, Al and Si was 1:1:1.9, which is in agreement with literature [34]. The biggest change of compressive strength with time happened when amount of added H_2_O was smallest and theoretical amount of ASN was biggest (see Figure 8b), i.e., when only Na-water glass was used, which has highest viscosity among all mixtures and therefore needs longest time for reaction to end and sample to dry. Drop of the compressive strength with time when mixture with ideal amount of substance ratio of the first group, Al and Si was used might be due to highest amount of added H_2_O, i.e., formation of more cracks while drying.

In Table 13 and Figure 9 is presented dependence of density and compressive strength of AAM on different mixtures close to literatures ideal amount of substance ratio of the first group, Al and Si [34] (see Figure 10a). Results are comparably high for all used mixtures, but showed a bit better results when there was a bit more liquid phase (see Figure 10b) than just minimal needed to work according to experiments.

### 3.4. Characteristics of Foamed AAM

After extensive testing of bulk AAM few mixtures have been selected for the foaming process. The most optimal mixture which was selected is ζ_0_^25^, which was used for all added foaming agents (Na-perborate, Al powder, H_2_O_2_).

#### 3.4.1. Foaming Agent Al

Compressive strength and density of AAF prepared with Al as foaming agent in powder form with different amounts of mixture ζ_0_ are presented in Table 14 and Figure 11. Amount of added Al was just 1%, because 5% led to over-foaming. Stabilizing agent was varied from 0.1%, 1% to 10%. From the results it can be concluded that to gain higher compressive strength it is better that there is less stabilizing agent, because in formation of AAF it positively influences only the surface tension and does not benefit the network formation. 

From results where the amount of Al is 1% and amount of Na-dodecyl sulphate is 1% it can be presumed that there is optimal value of added liquid phase, in this case 80% of mixture ζ_0_ calculated on the amount of precursor, resulting in density of 0.7 kg/l and compressive strength 2.8 MPa. Namely if there is not enough liquid phase not all precursor and foaming agent will react, and if there is too much liquid phase bubbles will be going out from the system until enough liquid phase will evaporate and strong enough ASN, to keep bubbles in the system, will form.

The optimal choice when ζ_0_ mixture and foaming agent Al were used is 40 g of mixture ζ_0_, 0.1% of Al and 1% of Na-dodecyl sulphate. Its SEM micrograph is presented in Figure 12. 

#### 3.4.2. Foaming Agent H_2_O_2_

Compressive strength and density of AAF prepared with H_2_O_2_ as foaming agent are presented in Table 15 and Figure 13 0.1%, 1%, and 10% of foaming agent and Na-dodecyl sulphate (as stabilizing agent) were used for AAF with mixture ζ_0_ in all possible combinations. As it was observed in experiments with Al, also when H_2_O_2_ was used as foaming agent, it is clear that for having higher compressive strength it is better to add less stabilizing agent. Optimal value of additives, resulting in density lower than 1 kg/l and compressive strength above 1 MPa, is 10% of H_2_O_2_ and 1% of Na-dodecyl sulphate; i.e., resulting in density 0.7 kg/l and compressive strength 4.6 MPa. 

Optimal choice when ζ_0_ mixture and foaming agent H_2_O_2_ were used is 25 g of mixture ζ_0_, 10% of H_2_O_2_, and 0.1% of Na-dodecyl sulphate. Its SEM micrograph is presented in Figure 14.

#### 3.4.3. Foaming Agent Na-Perborate

Compressive strength and density of AAF prepared with Na-perborate as foaming agent (powder) with different amounts of mixture ζ_4_ are presented in Table 16 and Figure 15. 

There is no significant dependence of density and compressive strength on the amount of mixture ζ_4_ observed when 1% of foaming and stabilizing agents are added, while bending strength is above zero only when minimal needed amount of mixture ζ_4_, i.e., 50%, is used (Table 16 and Figure 15), i.e., the mixture with ratio of amount of substance n_Al_:n_Si_:n_Na_ = 1:1:1.9 [34], which gives highest compressive strength with fly ash and is the most optimal result where density is below 1 kg/l, compressive strength above 1 MPa and has highest bending strength among comparable results. Optimal value of additives with 50% of mixture ζ_4_, resulting in density lower than 1 kg/l and compressive strength above 1 MPa, is 5% of Na-perborate and 1% of Na-dodecyl sulphate; i.e., resulting in density 0.8 kg/l and compressive strength 1.7 MPa. 

Compressive strength and density of AAF prepared with Na-perborate as foaming agent (powder) with different amounts of mixture ζ_0_ are presented in Table 17 and Figure 16.

No significant dependence of density and compressive strength on the amount of mixture ζ_0_ (Table 17 and Figure 16) was observed when 0.6% of foaming and stabilizing agents are added. With the addition of 0.1% of stabilizing agent and no foaming agent, density lowers for almost 20%, while compressive strength lowers for 90%. With the addition of 0.1% of stabilizing agent and 0.6% of foaming agent, density lowers for 45%, while compressive strength lowers for 60%. Density, bending and compressive strength lower with increasing amount of foaming and stabilizing agent. Optimal value of additives with 80% of mixture ζ_0_ according to the amount of precursor, resulting in density lower than 1 kg/l and compressive strength above 1 MPa, is 1% of Na-perborate and 1% of Na-dodecyl sulphate; i.e., resulting in density 0.7 kg/l and compressive strength 5.1 MPa. 

Difference between results of density, compressive, and bending strength is obvious from the SEM microstructure of bulk and foamed bulk presented in Figure 17, where from one to another sample from left to right only one parameter was slightly changed. The changes resulted in different distribution and shape of pores. While AAM had almost no pores (compared to foamed samples; micrograph in Figure 17a), density 1.9 kg/l, compressive strength 16.9 MPa, addition of 5% of foaming agent, and 1% of stabilizing agent (micrograph in Figure 17b) create large macro-pores longitudinally shaped lowering density to 0.3 kg/l and compressive strength to 0.12 MPa. Lowering the amount of foaming agent to 1% (micrograph in Figure 17c) reduces the amount of pores, but their shape stays longitudinal; density gets bigger (0.9 kg/l), so does compressive strength (2.5 MPa). Addition of liquid phase changed the amount of substance ratio (micrograph in Figure 17d) to too much Na according to the literature [34], lowered the density to 0.7 kg/l, kept the same value of the compressive strength, but lowered the bending strength from 1.3 to 0.05 MPa, due to the not ideal amount of substance ratio or due to the spherically shaped pores according to SEM micrograph. 

Optimal choice when ζ_4_ mixture and foaming agent Na-perborate were used is 25 g of mixture ζ_4_, 1% of Na-perborate, and 1% of Na-dodecyl sulphate. Its SEM micrograph is presented in Figure 18.

Optimal choice when ζ_0_ mixture and foaming agent Na-perborate were used is 40 g of mixture ζ_0_, 1% of Na-perborate, and 1% of Na-dodecyl sulphate. Its SEM micrograph is presented in Figure 19.

## 4. Conclusions

The potential of green ceramic waste was investigated for the use as precursor in alkali activation and synthesis of alkali activated foams. Thorough chemical and mineralogical analyses were performed, alkali activated materials and foams were synthesized, and their densities and compressive strengths were tested as most important final properties.

It was found that there are negative and positive effects influencing compressive strength that go on from the beginning of curing. The most important parameter for final mechanical strength is chemistry of the slurry, i.e., best compressive strength results were from samples where slurries had the ratio of amount of substance close to n_Al_:n_Si_:n_Na_ = 1:1.9:1 [34]. Samples with this ratio gained final compressive strength in short time no matter the temperature of curing, i.e., due to the chemical reactions ending soon (there were approximately “just enough elements” added for formation of the ASN).

The second most important parameter was viscosity of the alkali activator liquid, accompanied with the temperature of curing. Forced curing, i.e., curing at higher temperatures, especially accompanied with low viscosity of used liquid phase, results in gaining final compressive strength quicker, but compressive strength was lower and sample cracked, lower temperature, and higher viscosity of the used liquid result in slower reactions due to the hindered diffusion and slower dehydration of the samples, but final compressive strength is higher, samples with less cracks, just curing takes much more time.

Depending on conditions during alkali activation (chemical and physical) different crystals grow, some dissolve completely, some partially, but all effects contribute to the time formation of compressive strength and to the final compressive strength, meaning that the second approximation of additives in alkali activation should take into account elements that come from existing crystals in precursor, and elements that were used for growth of crystals especially crystals with lower hardness on Mohs scale.

Compressive strength and density decreased with the amount of foaming agents, even below measurable. Optimal results related to the demands that density should be as low as possible and compressive strength still above 1 MPa give the following results:With Al: mixture ζ_0_^40^, 0.1% of Al, 1% of Na-dodecylsulphate, ending with density 0.7 kg/l and compressive strength 2.8 MPa,With H_2_O_2_: mixture ζ_0_^25^, 1% of H_2_O_2_, 0.1% of Na-dodecylsulphate, ending with density 0.7 kg/l and compressive strength 4.6 MPa,With Na-perborate:
⚬Mixture ζ_0_^25^, 5% of Na-perborate, 0.1% of Na-dodecylsulphate, ending with density 0.8 kg/l and compressive strength 1.7 MPa,⚬mixture ζ_0_^40^, 0.6% of Na-perborate, 0.6% of Na-dodecylsulphate, ending with density 0.7 kg/l and compressive strength 5.1 MPa.

Density, bending, and compressive strength show high dependence on small changes in the amount of foaming agent and stabilizing agent, and smaller dependence on the amount of liquid phase.

## Figures and Tables

**Figure 1 materials-12-03563-f001:**
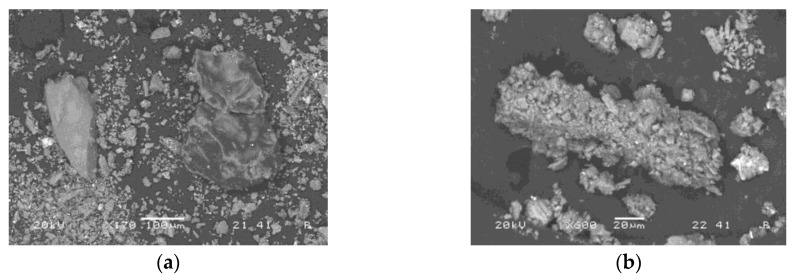
SEM micrographs of green waste ceramic. (**a**) General picture with rock-like particles; (**b**) aggregate.

**Figure 2 materials-12-03563-f002:**
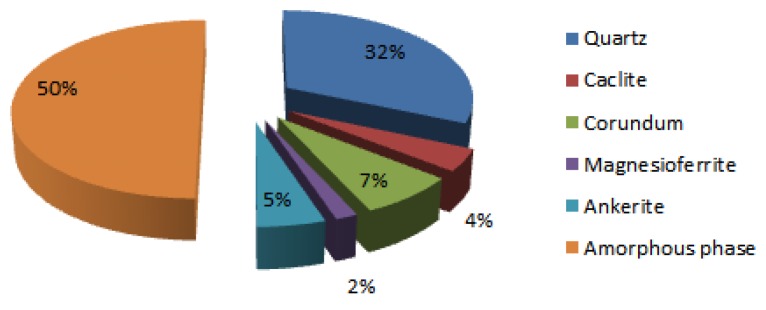
Mass percentage of minerals and amorphous phase in waste green ceramics determined from XRD measurement with Rietveld refinement with external standard.

**Figure 3 materials-12-03563-f003:**
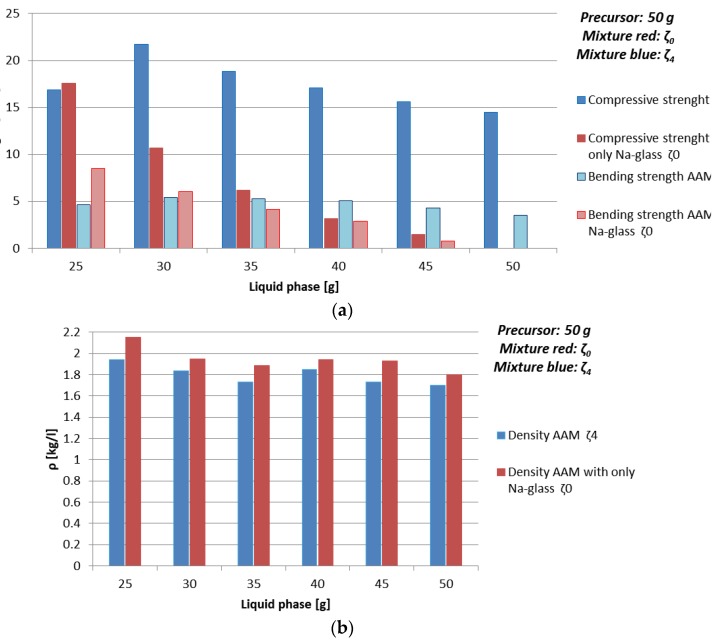
(**a**) Bending and compressive strength, and (**b**) density of AAM prepared with mixture ζ_0_ (red) and mixture ζ_4_ (blue), cured on 70 °C for one day, measured one week after molding.

**Figure 4 materials-12-03563-f004:**
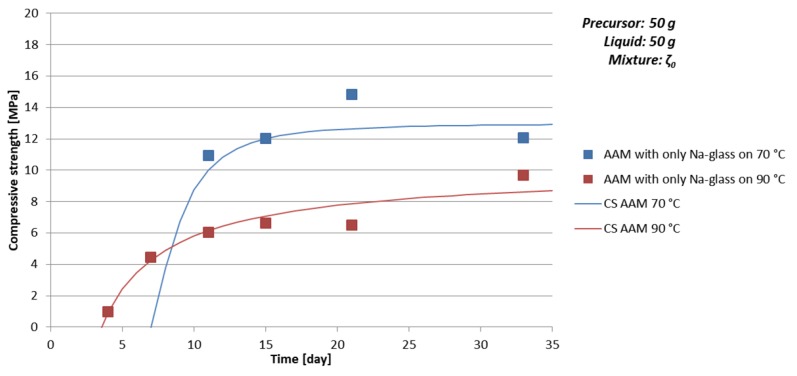
Time dependence of compressive strength of AAM prepared with mixture ζ_0_, cured on 70 °C (blue) for seven days and cured on 90 °C (red) for one day. Lines represent approximation of the compressive strength time development behaviour ${\mathrm{CS}}_{↑}$ (Equation (16)).

**Figure 5 materials-12-03563-f005:**
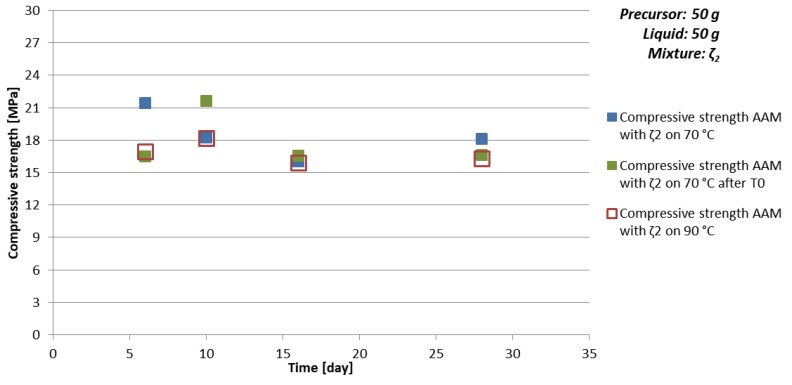
Time dependence of compressive strength of AAM prepared with mixture ζ_2_, cured on room temperature for 4 h and then on 70 °C for 20 h (**green**), cured on 70 °C for one day (**blue**), and cured on 90 °C for one day (**red**).

**Figure 6 materials-12-03563-f006:**
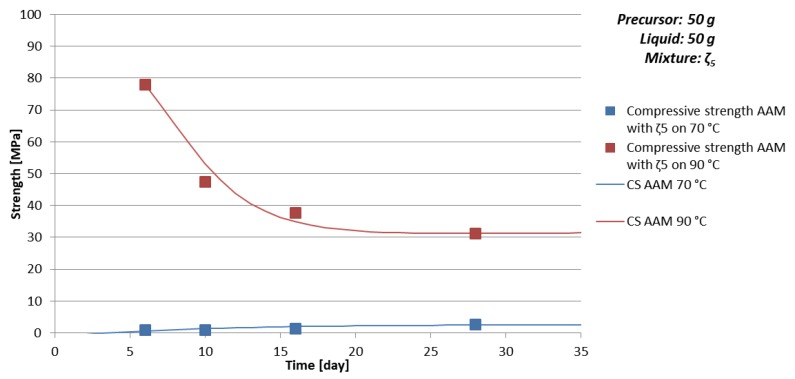
Time dependence of compressive strength of AAM prepared with mixture ζ_5_, cured on 70 °C (blue) for one day and cured on 90 °C (red) for one day. Lines represented approximation of the compressive strength time development behaviour $CS_{↑}$ (blue, Equation (16)) and $CS_{↓}$ (red, Equation (17)).

**Figure 7 materials-12-03563-f007:**
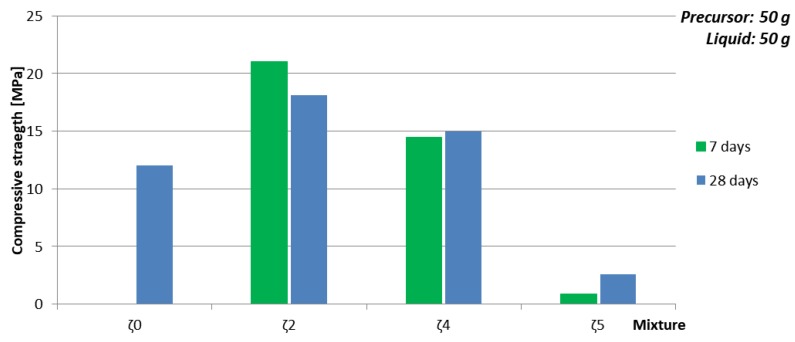
Compressive strength of AAM prepared with mixtures ζ_0_, ζ_2_, ζ_4_, and ζ_5_, cured on 70 °C for one day, measured one week after molding (green), and four weeks after molding (blue).

**Figure 8 materials-12-03563-f008:**
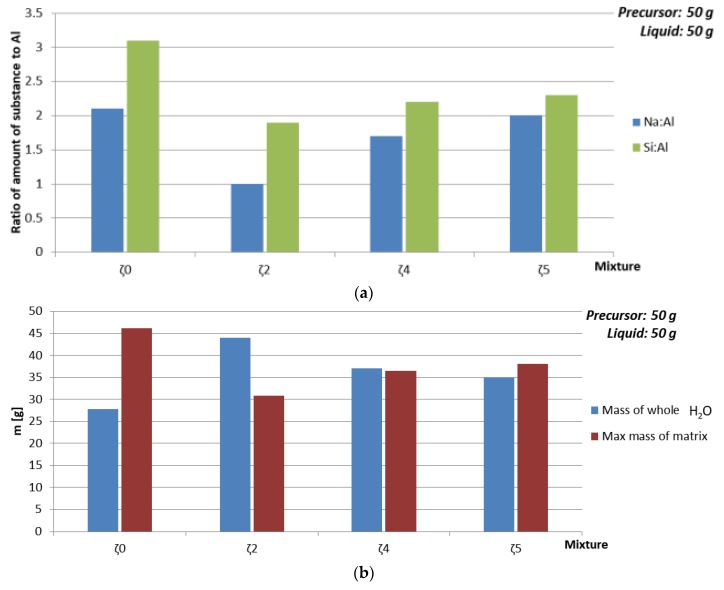
(**a**) Amount of substance according to amount of Al for Na and Si, (**b**) maximal mass of whole ASN and mass of whole water of AAM prepared with 50 g of mixtures ζ_0_, ζ_2_, ζ_4_, and ζ_5_, cured on 70 °C for one day.

**Figure 9 materials-12-03563-f009:**
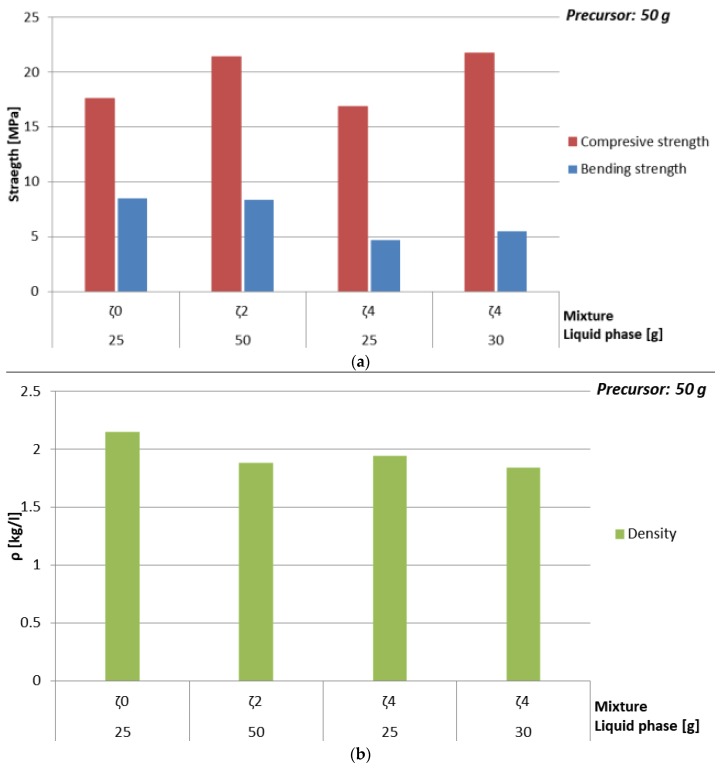
(**a**) Bending and compressive strength, and (**b**) density of AAM prepared with close to literature’s favourable amounts of substance ratios for the first group, Al and Si, cured on 70 °C for one day, measured one week after molding.

**Figure 10 materials-12-03563-f010:**
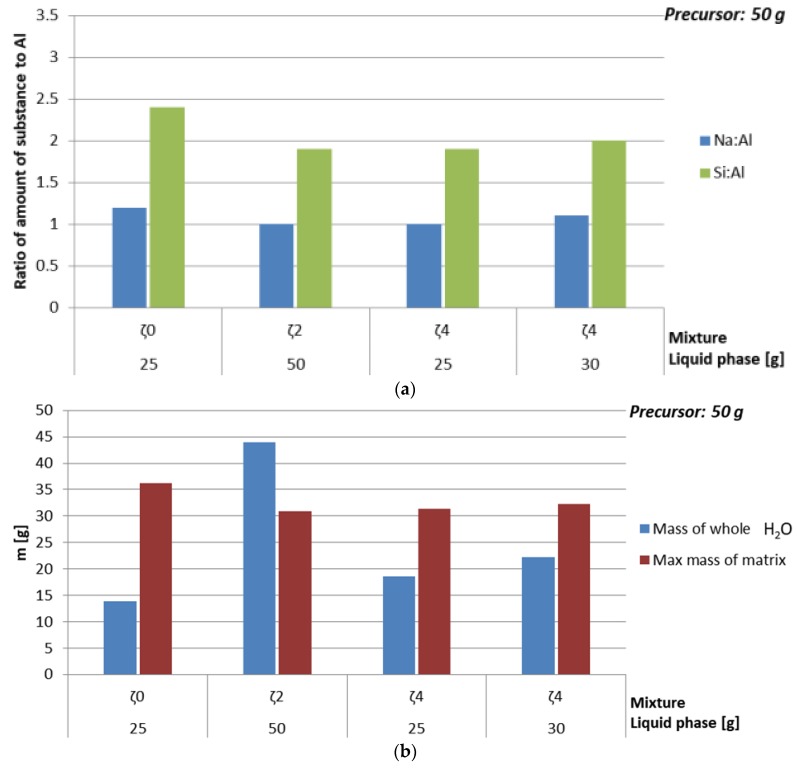
(**a**) Amount of substance according to the amount of Al for Na and Si, (**b**) maximal mass of whole ASN and mass of whole water of AAM prepared with close to literature’s favourable amounts of substance ratios for the first group, Al and Si, cured on 70 °C for one day, measured one week after molding.

**Figure 11 materials-12-03563-f011:**
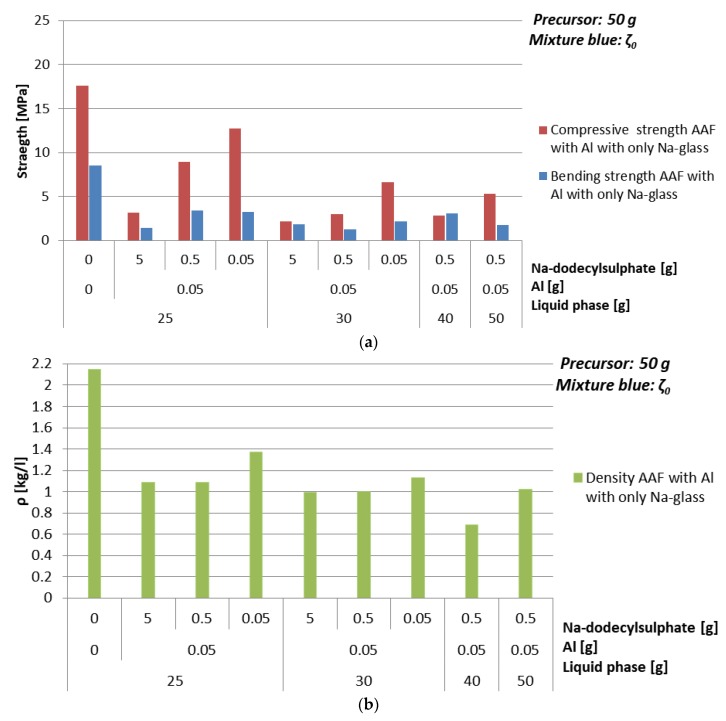
(**a**) Bending and compressive strength, and (**b**) density of AAF, produced by Al powder using mixture ζ_0_, cured on 70 °C for one day, measured one week after molding.

**Figure 12 materials-12-03563-f012:**
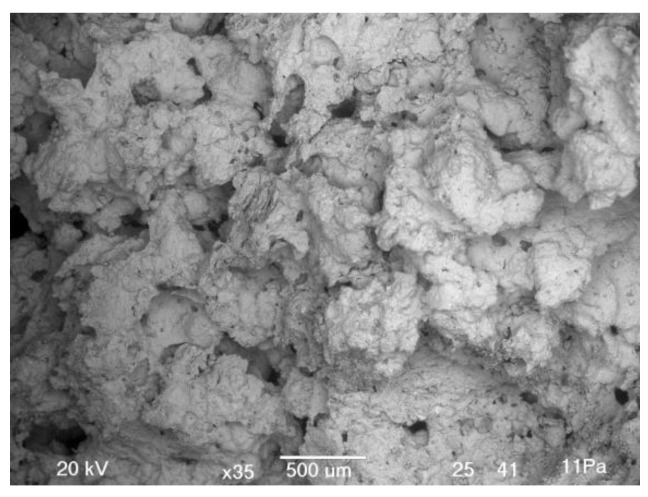
SEM micrograph of AAF prepared with 40 g of mixture ζ_0_, 50 g of precursor, 0.1% of Al, 1% of Na-dodecyl sulphate.

**Figure 13 materials-12-03563-f013:**
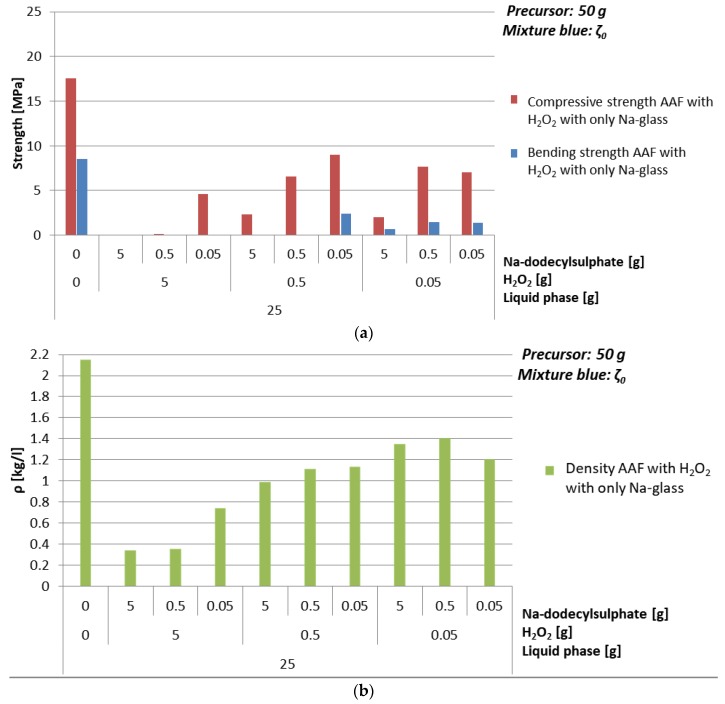
(**a**) Bending and compressive strength, (**b**) density of AAF, produced by H_2_O_2_ using 25 g of mixture ζ_0_, i.e., combination of ingredients for AA resulting in literature’s highest compressive strength, cured on 70 °C for one day, measured one week after molding.

**Figure 14 materials-12-03563-f014:**
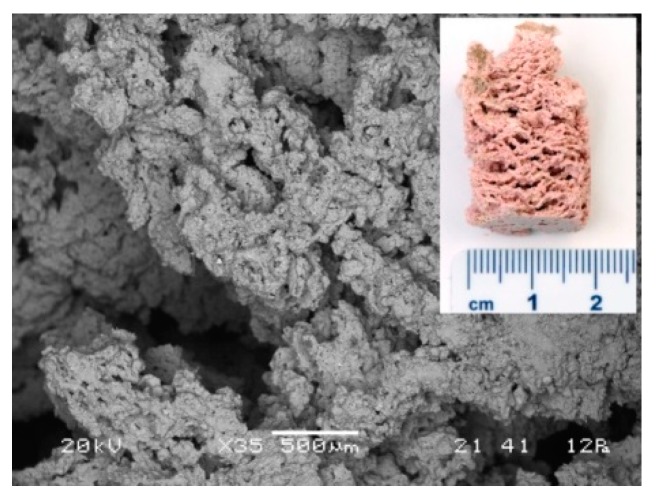
SEM micrograph of AAF prepared with 25 g of mixture ζ_0_, 50 g of precursor, 10% of H_2_O_2_, 0.1% of Na-dodecyl sulphate. On the micrograph’s inset is photography of cross-cut sample.

**Figure 15 materials-12-03563-f015:**
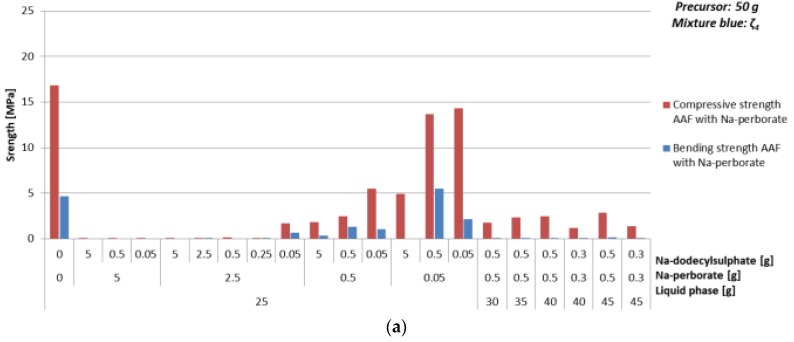
(**a**) Bending and compressive strength, and (**b**) density of AAF, produced by Na-perborate using mixture ζ_4_, cured on 70 °C for one day, measured one week after molding.

**Figure 16 materials-12-03563-f016:**
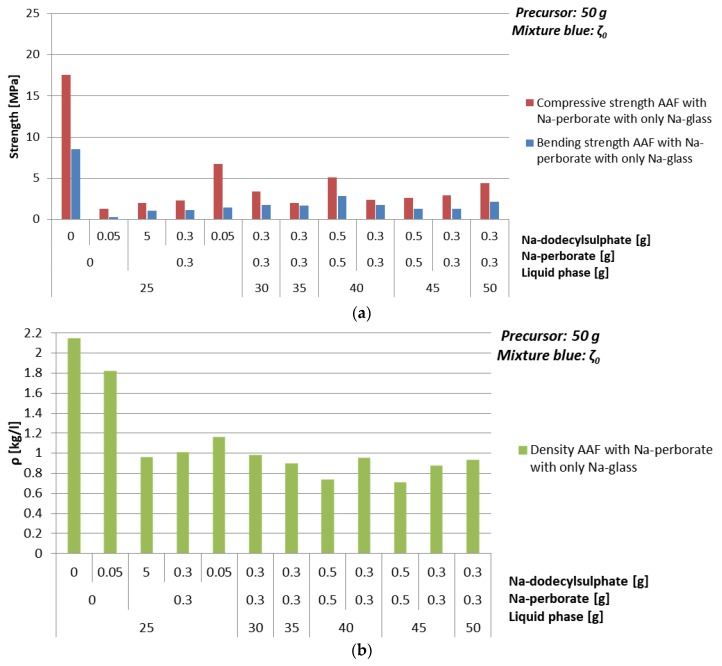
(**a**) Bending and compressive strength, and (**b**) density of AAF, produced by Na-perborate using mixture ζ_0_, cured on 70 °C for one day, measured one week after molding.

**Figure 17 materials-12-03563-f017:**
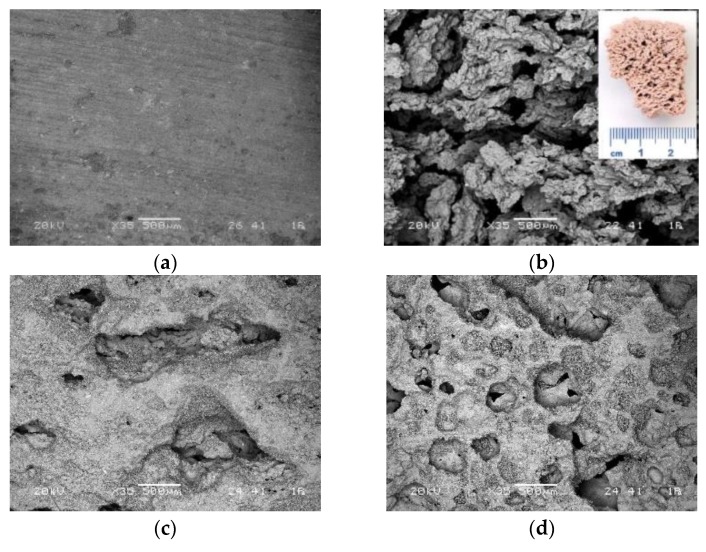
Dependence of the structure in SEM micrographs on small changes in preparation using mixture ζ_4_, Na-perborate as foaming agent and Na-dodecyl sulphate as stabilizing agent. (**a**) AAM (25 g of mixture ζ_4_), (**b**) AAF with 5% of foaming agent, and 1% of stabilizing agent (25 g of mixture ζ_4_, 2.5 g of foaming agent, 0.5 g of stabilizing agent; on the inset is photography of the cross-cut sample), (**c**) AAF with lower amount of foaming agent (25 g of mixture ζ_4_, 0.5 g of foaming agent, 0.5 g of stabilizing agent), (**d**) AAF with additional liquid phase (40 g of mixture ζ_4_, 0.5 g of foaming agent, 0.5 g of stabilizing agent). All samples are cured on 70 °C for one day and measured one week after molding.

**Figure 18 materials-12-03563-f018:**
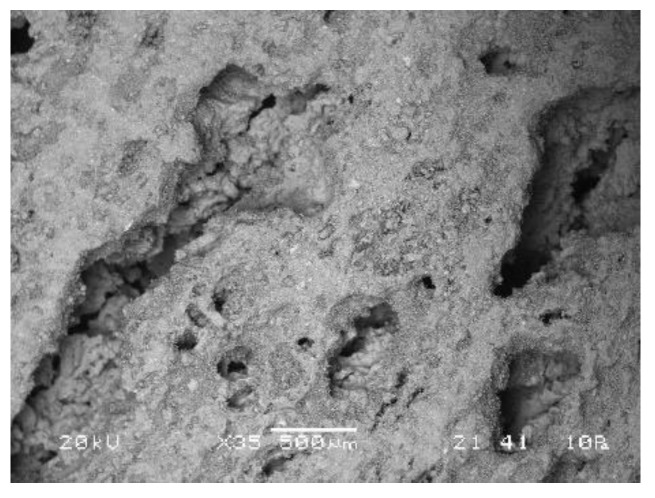
SEM micrograph of AAF prepared with 25 g of mixture ζ_4_, 50 g of precursor, 1% of Na-perborate, and 1% of Na-dodecyl sulphate.

**Figure 19 materials-12-03563-f019:**
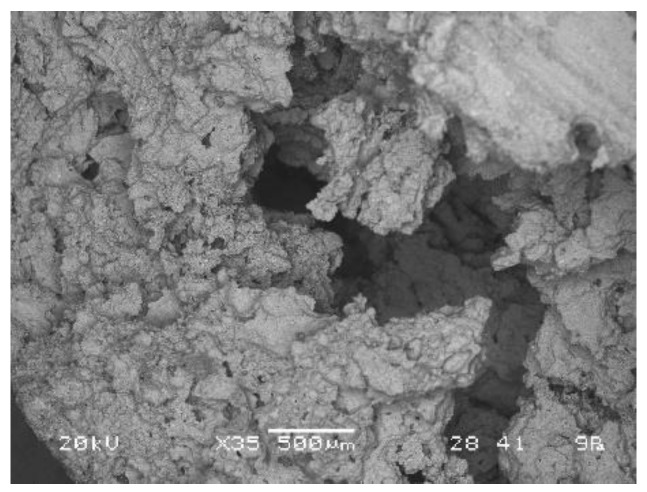
SEM micrograph of AAF prepared with 40 g of mixture ζ_0_, 50 g of precursor, 1% of Na-perborate, and 1% of Na-dodecyl sulphate.

**Table 1 materials-12-03563-t001:** Recipes for preparation of alkali activated materials with mass of all ingredients, i.e., Na, Si, and H_2_O. Combinations of ingredients for AA closest to literature’s ratio of amount of substance n_Al_:n_Si_:n_Na_ = 1:1:1.9 [34], that should give with fly ash the highest compressive strength and avoid efflorescence, are coloured. Mixtures are labeled with ζ_x_^y^, where x is amount of NaOH_(s)_ in 50 g of liquid phase, and y is amount of liquid used with 50 g of precursor.

**Mass Ratio [g]**									
**Mass Ratio NaOH: Na-water Glass:H_2_O = 0:50:0, labeled: mixture ζ_0_, viscosity = 0.1731 Pa·s**									
**Precursor (s)**	**NaOH (s)**		**Na-water Glass**				**Added Distilled H_2_O**	**Whole H_2_O**	**Whole Liquid Phase**
	**All Together**	**Na**	**All Together**	**Na/Na_2_O**	**Si/SiO_2_**	**H_2_O**			
50	0	0	25	3.1/4.2	3.2/6.8	13.9	0	13.9	25
50	0	0	30	3.8/5.1	3.9/8.3	16.7	0	16.7	30
50	0	0	35	4.4/5.9	4.5/9.6	19.5	0	19.5	35
50	0	0	40	5/6.7	5.1/10.9	22.2	0	22.2	40
50	0	0	45	5.8/7.8	5.8/12.4	25	0	25	45
50	0	0	50	6.3/8.5	6.4/13.7	27.8	0	27.8	50
**Mass Ratio NaOH: Na- water Glass:H_2_O = 2:9:39, labeled: mixture ζ_2_, viscosity = 0.02126 Pa·s**									
**Precursor (s)**	**NaOH (s)**		**Na-Water Glass**				**Added Distilled H_2_O**	**Whole H_2_O**	**Whole Liquid Phase**
	**All Together**	**Na**	**All Together**	**Na/Na_2_O**	**Si/SiO_2_**	**H_2_O**			
50	2	1.2	9	1.1/1.5	1.2/2.6	5	39	44	50
**Mass Ratio NaOH: Na-water Glass:H_2_O = 4:20:26, labeled: mixture ζ_4_, viscosity = 0.02402 Pa·s**									
**Precursor (s)**	**NaOH (s)**		**Na-water Glass**				**Added Distilled H_2_O**	**Whole H_2_O**	**Whole Liquid Phase**
	**All Together**	**Na**	**All Together**	**Na/Na_2_O**	**Si/SiO_2_**	**H_2_O**			
50	2	1.2	10	1.3/1.8	1.3/2.8	5.6	13	18.6	25
50	2.4	1.4	12	1.5/2.0	1.5/3.2	6.7	15.6	22.3	30
50	2.8	1.6	14	1.8/2.4	1.8/3.8	7.8	18.2	26	35
50	3.2	1.8	16	2/2.7	2.1/4.5	8.9	20.8	29.7	40
50	3.6	2.1	18	2.3/3.1	2.3/4.9	10	23.4	33.4	45
50	4	2.3	20	2.5/3.4	2.6/5.6	11.1	26	37.1	50
**Mass ratio NaOH: Na-water Glass:H_2_O = 5:22.5:22.5, labeled: mixture ζ_5_, viscosity = 0.02696 Pa·s**									
**Precursor (s)**	**NaOH (s)**		**Na-water Glass**				**Added Distilled H_2_O**	**Whole H_2_O**	**Whole Liquid Phase**
	**All Together**	**Na**	**All Together**	**Na/Na_2_O**	**Si/SiO_2_**	**H_2_O**			
50	5	2.9	22.5	2.8/3.8	2.9/6.2	12.5	22.5	35	50

**Table 2 materials-12-03563-t002:** Mass% of the foaming agents (Al, H_2_O_2_, Na-perborate), stabilizing agent (Na-dodecylsulphate), and precursor. Variations were used according to the success of experimental work mostly depending on the foaming agent effect.

Mass Ratio [g]		
Precursor	Foaming Agent	Na-dodecylsulphate
50	0	0, 0.5
	0.05	0.05, 0.5, 5
	0.3	0.03, 0.05, 0.3, 5
	0.5	0.05, 0.5, 5
	2.5	0.05, 0.25, 0.5, 2.5, 5
	5	0.05, 0.5, 5

**Table 3 materials-12-03563-t003:** Mass percentage of elements measured with XRF (m_%_^XRF^, first row), mass percentage of elements determined with Rietveld refinement from XRD (m_%_^XRD^, second row), mass percentage of elements in amorphous phase, i.e., elements useful for alkali activation (m_%_^amorphous^, third row).

Elements [m_%_]	Na	K	Cs	Mg	Ca	Sr	Ba	Al	Si
XRF	0.3	3.7	0	0.9	2.4	0.02	0.4	12.2	28.8
XRD	0	0	0	0.7	2.4	0	0	4.02	14.8
XRF-XRD	0.3	3.7	0	0.2	0	0.02	0.4	8.2	14.0

**Table 4 materials-12-03563-t004:** Mass ratio of precursor with useful material in the precursor for AA ASN formation (without and with O), NaOH, Na-water glass, and maximal value of ASN (if all useful material in precursor, whole Na from NaOH and Na-water glass, and whole Si from Na-water glass is used for ASN formation). Amount of substance ratio of amorphous first group, Al and Si according to the chosen mass of precursor, NaOH, Na-water glass. Combination of ingredients for AA resulting close to the literature’s highest compressive strength value [34] is coloured orange, initial value present in the precursor is coloured grey.

Mass Ratio [g]					Amount of Substance Ratio in ASN		
Precursor (s)	Useful Material for ASN in Precursor (without/with O)	NaOH (s)	Na- Water Glass (l)	ASN (Maximal Value)	1^st^ Group	Al	Si
50	13.3/26.1 (26.1 m_%_/52.3 m_%_)	0	0	26.1	0.3	1	1.6
		Mixture ζ_0_					
		0	25	36.2	1.2	1	2.4
		0	30	38.2	1.4	1	2.5
		0	35	40.2	1.6	1	2.7
		0	40	42.2	1.8	1	2.8
		0	45	44.2	2	1	3
		0	50	46.2	2.1	1	3.1
		Mixture ζ_2_					
		2	9	30.9	1	1	1.9
		Mixture ζ_4_					
		2	10	31.3	1	1	1.9
		2.4	12	32.3	1.1	1	2
		2.8	14	33.4	1.3	1	2.1
		3.2	16	34.4	1.4	1	2.1
		3.6	18	35.4	1.6	1	2.2
		4	20	36.5	1.7	1	2.2
		Mixture ζ_5_					
		5	22.5	38	2	1	2.3

**Table 5 materials-12-03563-t005:** Basic theoretically estimated parameters and measured mechanical properties of AAM synthesized using 50 g of mixtures ζ_0_, ζ_2_, and ζ_5_, cured first on room temperature (for 4 h) and then on 70 °C (for 20 h), and for 24 h on 70 and 90 °C.

Properties	Mixture, Temperature [°C]						
	ζ_0_		ζ_2_			ζ_5_	
	70 °C	90 °C	T_0_, 70 °C	70 °C	90 °C	70 °C	90 °C
Compressive strength after ~ 1 month [MPa]	12.1	9.7	16.6	18.1	16.3	2.6	31.2
Compressive strength’s time dependence (t)	grows	grows	const.	const.	const.	grows	falls
Mass of AAM in100 g [g]	72.2		56			65	
Mass of H_2_O in 100 g [g]	27.8		44			35	
Maximal mass of ASN [g]	46.2		30.9			38	
m_%_ of ASN [%]	64		55.2			58.5	
Amount of substance ratio 1^st^ group: Al:Si	2.1:1:3.1		1:1:1.9			2:1:2.3	

**Table 6 materials-12-03563-t006:** Rietveld refinement results (m_%_ of minerals and amorphous phase) of XRD measurements performed on precursor and one month old AAM synthesized using 50 g of mixtures ζ_0_, ζ_2_, and ζ_5_, cured first on room temperature (for 4 h) and then on 70 °C (for 20 h), and for 24 h on 70 and 90 °C. GOF is between 7.2 and 8.8.

Mineral				Mixture, Temperature [°C]						
				ζ_0_		ζ_2_			ζ_5_	
				70 °C	90 °C	T_0_, 70 °C	70 °C	90 °C	70 °C	90 °C
Geological Name	Chemical Formula	Mohs Hardness Scale [35]	Precursor m_%_ [%]	Mineral m_%_ [%]						
Corundum	Al_2_O_3_	9	7.6	21.4	21.9	29.9	28.2	28.8	23.6	24.6
Quartz	SiO_2_	7	31.8	5	3.6	9.5	7.6	7.2	6.9	6
Calcite	CaCO_3_	3	4.1							
Magnesioferrite	Fe_2_MgO_4_	6-6.5	1.7							
Ankerite	C_2_CaFe_0.23_Mg_0.77_O_6_	3.5-4	5.2							
Microcline maximum	AlK_0.95_Na_0.05_Si_3_O_8_	6-6.5		3.1						
Microcline intermediate	AlKSi_3_O_8_	6-6.5			15.1					
Orthoclase	AlKSi_3_O_8_	6		14.9	0.9	15.1	13.1	12.5	11.8	13.1
Zirconia	ZrO_2_	8-8.5		0.1	0.1	0.1	0.1		0.1	0.1
Kaolinite 2M	H_4_Al_2_Si_2_O_9_	2-2.5		9.5	8.5				10.5	1.9
Zeolite P1 (Na-exchanged)	H_24_Al_6_Na_6_Si_10_O_44_	4-5			1					
Zeolite SUZ-4	Al_5_K_5_O_72_Si_31_	4-5				0.8	0.7	0.8		
Albite low	AlNaSi_3_O_8_	6-6.5							2.7	
Albite (heat treated)	AlNaSi_3_O_8_	6-6.5			4.2			5.4		4.9
Sodalite MAPO-20	C_8_H_24_Al_4_Mg_2_N_2_O_24_P_6_	5.5-6				0.1	0.3		0.4	0.4
Amorphous [m_%_]		/	49.5	45.9	44.9	44.6	50	45.3	43.9	48.9

**Table 7 materials-12-03563-t007:** Scherrer crystallite sizes from XRD measurements performed on the precursor and one-month old AAM synthesized using 50 g of mixtures ζ_0_, ζ_2_, and ζ_5_, cured on room temperature followed by curing at 70, at 70 and 90 °C. Empty cells: Mineral is not present. “/”: Mineral is present, but crystallites are too big for valid determination of the diameter with Scherrer equation with X’Pert Highscore software (valid is up to 150 nm [36]).

Scherrer Size of Crystallites [nm]								
Mineral	Precursor	Mixture, Temperature [°C]						
		ζ_0_		ζ_2_			ζ_5_	
		70 °C	90 °C	T_0_, 70 °C	70 °C	90 °C	70 °C	90 °C
Corundum	130	100	100	100	95	100	90	80
Quartz	95	150	140	110	100	100	125	120
Calcite	65							
Magnesioferrite	/							
Ankerite	50							
Microcline maximum		100						
Microcline intermediate			20					
Orthoclase		5	/	25	20	30	15	10
Zirconia		/	/	/	/		/	/
Kaolinite 2M		20	20				15	/
Zeolite P1 (Na-exchanged)			/					
Zeolite SUZ-4				/	/	/		
Albite low							140	
Albite (heat treated)			60			70		45
Sodalite MAPO-20				/	/		/	/

**Table 8 materials-12-03563-t008:** Density, bending, and compressive strength of AAM prepared with mixture ζ_0_ and mixture ζ_4_, cured on 70 °C for one day, measured one week after molding. Combination of ingredients for AA resulting in literature’s highest compressive strength value is coloured.

Mixture	Whole Liquid Phase [g]	ρ [kg/l]	Bending Strength [MPa]	Compressive Strength [MPa]
ζ_0_	25	2.2	8.5	17.6
	30	2.0	6.0	10.7
	35	1.9	4.2	6.2
	40	1.9	2.9	3.2
	45	1.9	0.8	1.5
	50	1.8	0	0
ζ_4_	25	1.9	4.7	16.9
	30	1.8	5.5	21.7
	35	1.7	5.3	18.9
	40	1.9	5.1	17.1
	45	1.7	4.3	15.6
	50	1.7	3.5	14.5

**Table 9 materials-12-03563-t009:** Time dependence of density, bending, and compressive strength of AAM prepared with mixture ζ_0_, cured on 70 °C for seven days and cured on 90 °C for one day.

Mixture	Curing Temperature [°C]	Time [day]	ρ [kg/l]	Bending Strength [MPa]	Compressive Strength [MPa]
ζ_0_^50^	70	11	1.7	6.5	10.9
		15	1.8		12.0
		21	1.8	9.0	14.8
		33	1.9	8.4	12.1
	90	4	1.9	2.5	1.0
		7	1.9		4.5
		11	1.8	4.0	6.1
		15	1.8		6.6
		21	1.8	2.9	6.5
		33	1.7		9.7

**Table 10 materials-12-03563-t010:** Density, bending, and compressive strength of AAM prepared with mixture ζ_2_, having literature’s favourable amount of substance ratio of the first group: Al:Si [34], first cured on room temperature for 4 h and then on 70 °C for 20 h, cured on 70 °C for one day, and cured on 90 °C for one day.

Mixture	Curing Temperature [°C]	Time [day]	ρ [kg/l]	Bending Strength [MPa]	Compressive Strength [MPa]
ζ_2_^50^	T_0_, later 70	6	1.8	10. 5	16.5
		10	1.9		21.6
		16	2.0		16.5
		28	1.9		16.6
	70	6	1.9	8.4	21.4
		10	1.9		18.2
		16	1.9		16.0
		28	1.9		18.1
	90	6	1.6		17.0
		10	1.8		18.2
		16	1.7		15.9
		28	1.7	3.6	16.3

**Table 11 materials-12-03563-t011:** Time dependence of density, bending, and compressive strength of AAM prepared with mixture ζ_5_, cured on 70 °C for one day and cured on 90 °C for one day.

Mixture	Curing Temperature [°C]	Time [day]	ρ [kg/l]	Bending Strength [MPa]	Compressive Strength [MPa]
ζ_5_^50^	70	6	1.7	0.02	0.9
		10	1.7		0.9
		16	1.8	0.3	1.4
		28	1.7		2.6
	90	6	1.8	11.5	78.0
		10	1.8	13.8	47.4
		16	1.7	12.0	37.7
		28	1.8	11.1	31.2

**Table 12 materials-12-03563-t012:** Density, bending, and compressive strength of AAM prepared with same amount of liquid phase and different amount of substance ratios of first group, Al and Si, cured on 70 °C for one day, measured one week after molding, and compressive strength four weeks after molding. Combination of ingredients for AA resulting in literature’s highest compressive strength value is coloured.

Mixture	Viscosity [Pa·s]	Mass of Whole H_2_O [g]	Max Mass of ASN [g]	Amount of SubstanceNa:Al:Si	ρ 7 days [kg/l]	Bending Strength 7 days [MPa]	Compressive Strength 7 days [MPa]	Compressive Strength 28 days [MPa]
ζ_0_^50^	0.1731	27.8	46.2	2.1:1:3.1	1.7	0	0	12.01
ζ_2_^50^	0.02126	44	30.9	1:1:1.9	1.9	8.35	21	18.14
ζ_4_^50^	0.02402	37.1	36.5	1.7:1:2.2	1.7	3.5	14.5	15
ζ_5_^50^	0.02696	35	38	2:1:2.3	1.7	0.015	0.9	2.55

**Table 13 materials-12-03563-t013:** Density, bending, and compressive strength of AAM prepared with close to literature’s favourable amounts of substance ratios for the first group, Al and Si [34], cured on 70 °C for one day, measured one week after molding.

Mixture	Mass of Whole H_2_O [g]	Max Mass of ASN [g]	Amount of Substance Na:Al:Si	ρ [kg/l]	Bending Strength [MPa]	Compressive Strength [MPa]
ζ_0_^25^	13.9	36.2	1.2:1:2.4	2.2	8.5	17.6
ζ_2_^50^	44.0	30.9	1:1:1.9	1.9	8.4	21.4
ζ_4_^25^	18.6	31.3	1:1:1.9	1.9	4.7	16.9
ζ_4_^30^	22.3	32.3	1.1:1:2	1.8	5.5	21.7

**Table 14 materials-12-03563-t014:** Density, bending, and compressive strength of AAF, produced by Al powder using mixture ζ_0_, cured at 70 °C for one day, measured one week after molding. Combination of ingredients for AA resulting in literature’s highest compressive strength value is coloured.

Mixture	Al [g]	Na-dodecyl Sulphate [g]	ρ [kg/l]	Bending Strength [MPa]	Compressive Strength [MPa]
ζ_0_^25^	0	0	2.2	8.5	17.6
	0.05	5	1.1	1.4	3.2
		0.5	1.1	3.4	8.9
		0.05	1.4	3.2	12.8
ζ_0_^30^	0.05	5	1.0	1.8	2.2
		0.5	1.0	1.2	3.0
		0.05	1.1	2.1	6.6
ζ_0_^40^	0.05	0.5	0.7	3.1	2.8
ζ_0_^50^	0.05	0.5	1.0	1.8	5.3

**Table 15 materials-12-03563-t015:** Density, bending, and compressive strength of AAF, produced by H_2_O_2_ using 25 g of mixture ζ_0_, i.e., combination of ingredients for AA resulting in literature’s highest compressive strength, cured on 70 °C for one day, measured one week after molding.

Mixture	H_2_O_2_ [g]	Na-dodecyl Sulphate [g]	ρ [kg/l]	Bending Strength [MPa]	Compressive Strength [MPa]
ζ_0_^25^	0	0	2.2	8.5	17.6
	5	5	0.3	0.01	0.04
		0.5	0.4	0.04	0.1
		0.05	0.7	0	4.6
	0.5	5	1.0	0.04	2.3
		0.5	1.1	0.005	6.6
		0.05	1.1	2.4	9.0
	0.05	5	1.4	0.6	2.0
		0.5	1.4	1.4	7.6
		0.05	1.2	1.4	7.0

**Table 16 materials-12-03563-t016:** Density, bending, and compressive strength of AAF, produced by Na-perborate using mixture ζ_4_, cured on 70 °C for one day, measured one week after molding. Combination of ingredients for AA resulting in literature’s highest compressive strength value is coloured.

Mixture	Na-perborate [g]	Na-dodecyl Sulphate [g]	ρ [kg/l]	Bending Strength [MPa]			Compressive Strength [MPa]
ζ_4_^25^	0	0	1.9	4.7			16.9
	5	5	0.4			0	0.07
		0.5	0.4			0	0.06
		0.05	0.4			0	0.06
	2.5	5	0.3			0	0.02
		2.5	0.4			0.07	0.07
		0.5	0.3			0	0.1
		0.25	0.4			0.07	0.06
		0.05	0.8			0.7	1.7
	0.5	5	0.9			0.3	1.8
		0.5	0.9			1.3	2.5
		0.05	1.2			1.0	5.5
	0.05	5	1.4			0	4.9
		0.5	1.7			5.5	13.7
		0.05	1.7			2.2	14.3
ζ_4_^30^	0.5	0.5	0.8			0.02	1.8
ζ_4_^35^	0.5	0.5	0.7			0.02	2.4
ζ_4_^40^	0.5	0.5	0.7			0.05	2.5
	0.3	0.3	0.8		0.05		1.2
ζ_4_^45^	0.5	0.5	0.8		0.1		2.9
	0.3	0.3	0.8		0.02		1.4

**Table 17 materials-12-03563-t017:** Density, bending, and compressive strength of AAF, produced by Na-perborate using mixture ζ_0_, cured on 70 °C for one day, measured one week after molding. Combination of ingredients for AA resulting in literature’s highest compressive strength value is coloured.

Mixture	Na-perborate [g]	Na-dodecyl Sulphate [g]	ρ [kg/l]	Bending Strength [MPa]	Compressive Strength [MPa]
ζ_0_^25^	0	0	2.2	8.5	17.6
		0.05	1.8	0.3	1.3
	0.3	5	1.0	1.0	2.0
		0.3	1.0	1.1	2.3
		0.05	1.2	1.4	6.7
ζ_0_^30^	0.3	0.3	1.0	1.7	3.4
ζ_0_^35^	0.3	0.3	0.9	1.7	2.0
ζ_0_^40^	0.5	0.5	0.7	2.8	5.1
	0.3	0.3	1.0	1.7	2.4
ζ_0_^45^	0.5	0.5	0.7	1.3	2.6
	0.3	0.3	0.9	1.3	2.9
ζ_0_^50^	0.3	0.3	0.9	2.1	4.4

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
