# Peer review of "Potential of Green Ceramics Waste for Alkali Activated Foams"

_materials, 2019, doi:10.3390/ma12213563_

Round 1
Reviewer 1 Report
The manuscript, entitled “Potential of Green Ceramics Waste for Alkali Activated Foams”, with the number 590449, presents the results of an experimental study on the use of waste green body, used for ceramics’ production, as precursor for foamed alkali activated materials using different foaming agents. The investigation itself is worthy of publication as the concept is innovative and well developed throughout the manuscript. However, several issues were identified throughout the manuscript, one of which is the lack of summarization of information; the manuscript is simply too long and gives off the sense of lack of direction. The information should be further summarized in a way that it does not interfere with the contents. Furthermore, given the amount of information, it is highly recommended that it is divided and published in two papers, rather than a very long one.
When the authors state green ceramics, they are actually mentioning the green body prior to thermal processing. The classification 10 12 01, which the authors say is from the list of waste from Official Gazette of the Republic of Slovenia, no. 20/01 Annex 1, actually it is originally from the European Waste Catalogue, and subsequently absorbed by Slovenia for local legislation.
The text is somewhat well written but contains several syntax errors. The authors should pay closer attention to the writing of the manuscript. This reviewer suggests submitting the manuscript to an English-speaking native. Although conciseness is very well appreciated, the text is extremely robotic and lacks fluidity. Several issues were identified including:
Fly ash does not have a hyphen. There are several chemical notations in which the numbers are not subscripted. The construction of many sentences does not have a correct use of commas. When citing, it is not “et.al.” but “et al.”. Please change all applicable cases. Construction of sentences where words were eaten. For example, “decrease happened in first 7 days” should be “decrease happened in the first 7 days”, “even bigger decrease of compressive strength is with increase of amount of H2O2” should be “even bigger decrease of compressive strength occurred with increasing amount of H2O2”, “Lowest compressive strength, 0.26 MPa, that also had lowest density, 0.44 kg/l, was obtained with 8 M NaOH, more water and more H2O2.” goes without saying and this was all in one single paragraph. There are some instances in the manuscript that give off the impression that most was copied from another document with little regard to the quality of the final paper – “(in amount of amorphous phase and compressive strength; see this chapter below)”. Naturally, there are no chapters in this document.Is the compressive strength of last mix in table 5 correct (31.2 MPa)? There should not be such a big difference between mixes cured at 70 ºC and 90 ºC.
Line 423 and in other places – Things can only drop downwards. Please, correct the redundancy.
The regressions in Figure 4 and Figure 6 are unrepresentative. Please, delete them and the corresponding text. Additional, to avoid such a confusion of information, it would be best to divide the properties in different graphs.
Table 12 has two compressive strength columns, but only one of them is explicit in terms of age.
Figure 8 could be transformed into a column’s graph. Same with Figure 9 and 10.
Reviewer 2 Report
Very good systematic study, leading to optimal foamed materials
Language needs extensive revision. Please consider review by a native English speaker or language specialist.
Introduction
Content is good and extensive on the topic of alkali-activation and alkali-activated foams. It can be written more efficiently when some studies are summarized together instead of discussing every study separately.
Some details about the current use of residual green ceramic would be interesting.
Materials and methods
Methods and material are in general properly described and defined. Some missing details which should be reported are:
Voltage and current used during XRF and XRD Preparation of SEM samples (grinding, usually polishing but figure 9, 12 and 22 show this might have been skipped, normally also coating but this might not be necessary in the low vacuum), conditions during the EDS measurements (energy/acceleration voltage), optionally, the software of the EDS can be added.
There is overlap between section 2.1 and 2.4: you can consider merging them into 1 characterization section?
In section 2.2, it is difficult to follow table with mixtures. Maybe it would be better if the components are presented in oxides instead of elements? I feel this is more common in literature.
Results and discussion
Figure 10: I have doubts of the usefulness of plotting the Al/Al ratio.
SEM micrographs: Polishing is badly or not done, grinding lines are still clearly visible (at least in Figure 9, 12 and 22). These should be redone or the EDS can be omitted from the paper as it does not contribute significantly to the story or is not needed to support the main conclusions of the paper.
Very good systematic study, leading to optimal foamed materials
Language needs extensive revision. Please consider review by a native English speaker or language specialist.
Introduction
Content is good and extensive on the topic of alkali-activation and alkali-activated foams. It can be written more efficiently when some studies are summarized together instead of discussing every study separately.
Some details about the current use of residual green ceramic would be interesting.
Materials and methods
Methods and material are in general properly described and defined. Some missing details which should be reported are:
Voltage and current used during XRF and XRD Preparation of SEM samples (grinding, usually polishing but figure 9, 12 and 22 show this might have been skipped, normally also coating but this might not be necessary in the low vacuum), conditions during the EDS measurements (energy/acceleration voltage), optionally, the software of the EDS can be added.
There is overlap between section 2.1 and 2.4: you can consider merging them into 1 characterization section?
In section 2.2, it is difficult to follow table with mixtures. Maybe it would be better if the components are presented in oxides instead of elements? I feel this is more common in literature.
Results and discussion
Figure 10: I have doubts of the usefulness of plotting the Al/Al ratio.
SEM micrographs: Polishing is badly or not done, grinding lines are still clearly visible (at least in Figure 9, 12 and 22). These should be redone or the EDS can be omitted from the paper as it does not contribute significantly to the story or is not needed to support the main conclusions of the paper.

Round 2
Reviewer 1 Report
Assuming that the authors will submit the manuscript to the MDPI English Editing Service, I agree with the changes and can now be accepted for publication.